# AutoDrive-R²: Incentivizing Reasoning and Self-Reflection Capacity for VLA Model in Autonomous Driving

**Zhenlong Yuan**[1][*][†]**, Chengxuan Qian**[1][*]**, Jing Tang**[1]**, Rui Chen**[1]**, Zijian Song**[2]**,
**Lei Sun**[1][‡]**, Xiangxiang Chu**[1]**, Yujun Cai**[3]**, Dapeng Zhang**[4][§]**, Shuo Li**[5]

[1] AMAP, Alibaba Group, [2] Sun Yat-sen University, [3] University of Queensland,
[4] Lanzhou University, [5] Case Western Reserve University

[*] Equal contribution   [‡] Project Lead   [§] Corresponding Author

## Abstract

Vision–Language–Action (VLA) models in autonomous driving systems have recently demonstrated transformative potential by integrating multimodal perception with decision-making capabilities. However, the interpretability and coherence of the decision process and the plausibility of action sequences remain largely underexplored. To address these issues, we propose AutoDrive-R², a novel VLA framework that enhances both reasoning and self-reflection capabilities of autonomous driving systems through chain-of-thought (CoT) processing and reinforcement learning (RL). Specifically, we first propose an innovative CoT dataset named nuScenesR²-6K for supervised fine-tuning, which effectively builds cognitive bridges between input information and output trajectories through a four-step logical chain with self-reflection for validation. Moreover, to maximize both reasoning and self-reflection during the RL stage, we further employ the Group Relative Policy Optimization algorithm within a physics-grounded reward framework that incorporates spatial alignment, vehicle dynamic, and temporal smoothness criteria to ensure reliable and realistic trajectory planning. Extensive evaluation results across both nuScenes and Waymo datasets demonstrates the state-of-the-art performance and robust generalization capacity of our method.

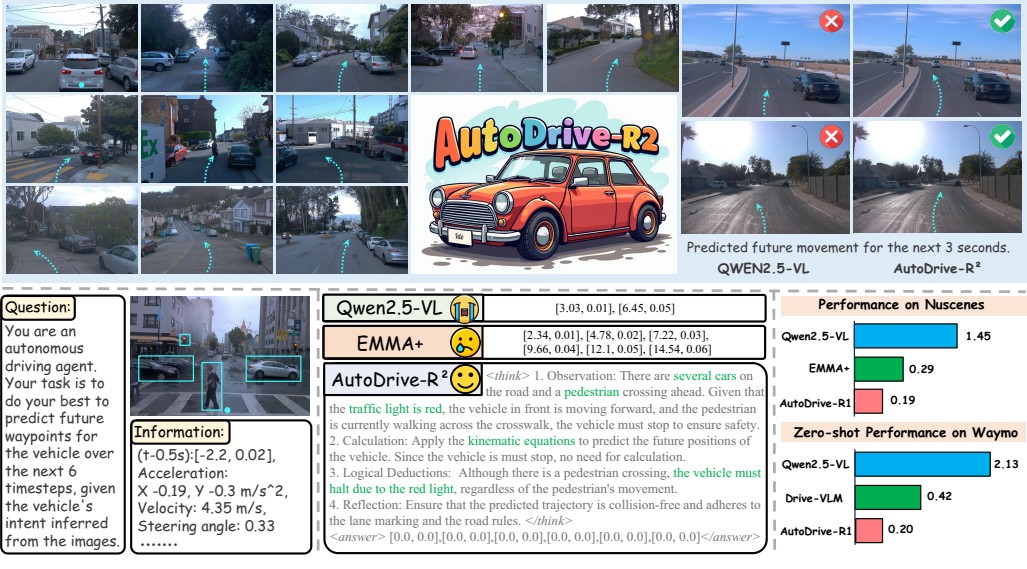

---

[†]Work done during the internship at AMAP, Alibaba Group.

## 1 INTRODUCTION

Autonomous driving technologies have witnessed rapid advancements in recent years. These systems typically take sensor data as input and then output planning trajectories. Traditional pipelines (Kendall et al., 2019; Chen et al., 2021) usually adopt architectures with separate perception, mapping, prediction, and planning modules. Such design may suffer from two key limitations: error accumulation and lack joint optimization across components, leading to performance degradation. In contrast, modern methods (Hu et al., 2023; Jiang et al., 2023; Chen et al., 2024) unify these complex systems into a single end-to-end paradigm, which naturally offers three main benefits: system simplification, enhanced robustness, and alleviated error accumulation.

However, these end-to-end methods primarily focus on trajectory planning while lacking the contextual reasoning necessary for complex driving scenarios. To address this limitation, recent works integrate Vision-Language Models (VLMs) into autonomous driving systems, leveraging their pretrained reasoning capabilities to enhance decision-making in challenging situations (Shao et al., 2024a; Wang et al., 2023; Tian et al., 2024). Unlike traditional approaches that train perception-policy modules from scratch, VLM-based methods instead fine-tune pre-trained models by leveraging pre-training on millions of image-text pairs, enhancing vehicles to interpret dynamic traffic situations and develop sophisticated navigation strategies. Despite promising results, current VLM-based systems still struggle with consistently producing accurate planning outputs.

Building upon VLMs, Vision-Language-Action models (VLA) further extend reasoning capabilities to final action prediction, enabling robots and autonomous vehicles to generate precise actions from visual inputs and textual instructions (Yang et al., 2025). This advancement has led to the adoption of similar action generation mechanisms in autonomous driving, with approaches like $\pi 0$ (Black et al., 2024) inspiring the development of action tokenizer that produce precise planning trajectories (Zhou et al., 2025b).

However, current VLA approaches in autonomous driving typically face two critical limitations that hinder their practical deployment: First, existing trajectory generation framework often produce physically infeasible outputs. Existing approaches that directly generate textual commands or waypoints via VLMs frequently produce physically-infeasible outputs and exhibit model collapse. While intermediate representations like meta-actions or latent action tokens have been proposed to mitigate these issues, these designs violate the end-to-end optimization principle and significantly increase model complexity overhead. Second, current systems demonstrate inadequate reasoning capabilities for complex driving scenarios. Since most methods employ simplistic reasoning strategies, they fail to account for both complicated road condition and vehicle kinematic constraints, resulting predicted trajectories significantly deviate from real-world requirements. These limitations underscore the critical need for a novel VLA framework that balances architectural simplicity, robust contextual understanding, and strict physical constraints.

To overcome these challenges, we propose AutoDrive-R², a novel VLA framework that enhances both reasoning quality and physical feasibility through a two-stage training approach. Our key insight is that effective autonomous driving requires structured reasoning processes that can be systematically validated and refined. Specifically, to address the inadequacy of contextual reasoning for complex driving scenarios, we first construct nuScenesR²-6K, a chain-of-thought (CoT) dataset for supervised fine-tuning (SFT). nuScenesR²-6K is the first dataset in autonomous driving that stimulates both reasoning and self-reflection capabilities for VLA models. Unlike prior datasets, nuScenesR²-6K provides not only ground-truth trajectories but also the underlying reasoning and self-reflection steps, ensuring both the correctness and causal plausibility of driving behavior.

Furthermore, to resolve the challenge of physically infeasible trajectory generation, we further develop a physics-grounded reward framework tailored to group relative policy optimization (GRPO) of autonomous driving tasks. By explicitly incorporating spatial alignment, vehicle dynamic and temporal smoothness constraints into consideration, our physics-grounded reward enables reinforcement learning to adapt to diverse driving scenarios and vehicle dynamics while maintaining physical feasibility and motion comfort. Comprehensive experiments on nuScenes and Waymo benchmarks demonstrate that AutoDrive-R² achieves state-of-the-art performance. Our key contributions are:

- We introduce AutoDrive-R², a novel VLA framework that enables semantic reasoning with self-reflection step and trajectory planning from visual information and language instructions.

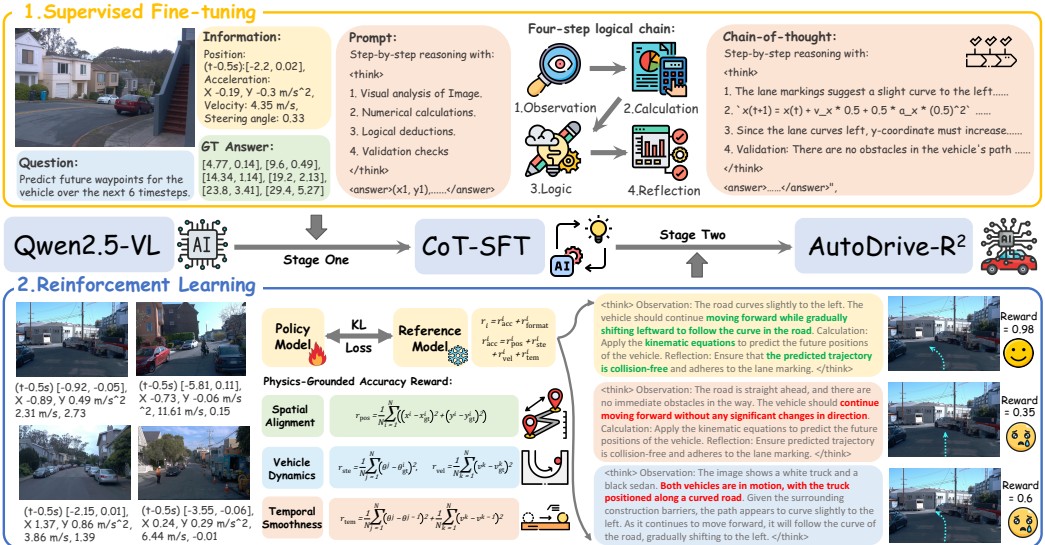

Figure 1: Pipeline of our method. We adopt a two-stage training process. The first stage introduces an innovative CoT dataset named nuScenesR²-6K for SFT. The nuScenesR²-6K adopts a four-step logical chain with self-reflection to generate valuable chain-of-thought data. The second stage proposes an novel physics-grounded reward framework for RL optimization, which incorporates spatial alignment, vehicle dynamic, and temporal smoothness for reliable trajectory planning.

- We propose nuScenesR²-6K, an innovative CoT dataset adopting a four-step logic chain with self-reflection to help build foundational perception capabilities after SFT.
- We propose a RL-based post-training method based on GRPO, which incorporates physics-grounded rewards as constraint to refine planning trajectory for diverse scenes.

## 2 METHODOLOGY

**Overview.** In this section, we present an overview of AutoDrive-R². The target of trajectory planning task requires the model to forecast a vehicle's future motion based on its historical sensor data and contextual information. Formally, given a sequence of historical vehicle states $H$ (including position, acceleration, velocity, steering angle, etc.) and its camera image $F$, the model $M$ outputs predicted bird's-eye view (BEV) trajectory coordinates $T$ over the next 3 seconds at 0.5-second intervals, defined as $T = M(H, F)$.

As depicted in Fig. 2, our training process contains two stages. In stage one, we construct a high-quality dataset nuScenesR²-6K for cold start to build cognitive bridges between input information and output trajectories through a four-step logical chain with self-reflection for validation. In stage two, we employ a physics-based reinforcement learning framework that integrates spatial alignment, vehicle dynamic and temporal smoothness to ensure physically feasible trajectory generation.

### 2.1 LOGISTIC COT DATASET WITH SELF-REFLECTION

The success of VLA models in autonomous driving critically depends on their ability to produce both interpretable reasoning and physically feasible actions. However, existing training approaches often fail to establish this dual requirement, leading to models that either lack explainable decision-making processes or generate unrealistic trajectories. To investigate this challenge, we initially explored direct reinforcement learning optimization for trajectory planning, following recent advances in reasoning-based RL (Guo et al., 2025). However, preliminary experiments revealed that models trained exclusively on RL exhibited significant degradation in trajectory planning compared to models pre-trained with SFT before RL. Therefore, we constructed a high-quality cold-start dataset named nuScenesR²-6K to cultivate the model's foundational understanding of trajectory planning.

To this end, we constructed nuScenesR²-6K, a dataset of 6,000 image-trajectory pairs enriched with high-fidelity chain-of-thought (CoT) reasoning. The dataset was created through a meticulous "generate-then-validate" pipeline. First, we curated an initial pool of approximately 8,000 image-trajectory pairs from the nuScenes training set. We then leveraged the Qwen2.5-VL-72B model to synthesize initial CoT reasoning sequences for these pairs. Crucially, to ensure data quality, we employed the closed-source Qwen-VL-Max model as an expert validator to score and review each generated chain. Chains containing factual errors or logical inconsistencies were systematically discarded. This rigorous filtering process yielded the final 6,000 high-fidelity samples for SFT.

Moreover, we observe that many existing approaches rely on universal prompts for problem-to-answer reasoning, lacking structured guidance for rational analysis. While this strategy proves effective for simple tasks, it frequently fails when confronted with complex mathematical or logical problems. To address this limitation, our CoT prompt design systematically decomposes trajectory planning into three interdependent reasoning stages:

- Image-Driven Analysis: Establishing foundational scene understanding (e.g., obstacle and lane localization, traffic sign detection) to anchor subsequent reasoning.
- Physics-based Calculation: Leveraging kinematic equations (e.g., angular momentum conservation) to translate abstract observations into quantifiable predictions.
- Contextual Logic Synthesis: Integrating domain-specific knowledge (e.g., intersection traffic rules) to ensure predictions align with real-world driving regulations.

To further enhance robustness and the correctness of answers, we explicitly introduce a self-reflection phase as the fourth step, inspired by mathematical reasoning frameworks that validate conclusions through backward-checking. This allows the model to verify the coherence of its reasoning and correct potential contradictions. Our prompt implements a four-step logic chain:

$$\textbf{Observation} \rightarrow \textbf{Calculation} \rightarrow \textbf{Logic} \rightarrow \textbf{Reflection,}$$

which delivers both systematic and error-resilient reasoning. Ultimately, the nuScenesR²-6K dataset is adopted for supervised fine-tuning Qwen2-VL-7B model, thus yielding our stage-1 model.

## 2.2 GROUP RELATIVE POLICY OPTIMIZATION (GRPO)

We follow the GRPO algorithm (Jiang et al., 2025) to train the model. Unlike traditional approaches that rely on critic networks to estimate value functions, GRPO introduces a pairwise comparison mechanism among candidate response. This design not only simplifies the architecture but also reduces computational overhead during training. The methodology begins by generating $G$ distinct candidate responses $o = \{o_1, \ldots, o_G\}$ for a given input question $q$ through policy sampling. For our specific task, we implement two rule-based verifiable reward functions to assess response quality:

**Accuracy Reward** To better adapt to trajectory planning task, we define a physics-grounded accuracy rewards $r_{acc}$ which integrates spatial, kinematic, and temporal constraints for evaluation. Details are specified in the following section.

**Format Reward** The format reward $r_{acc}$ enforces strict adherence to the required output format. The model must produce responses in the form: *"<think>thinking process here</think><answer>$(x_1, y_1)$, ...,$(x_n, y_n)$</answer>"*. A value of 1 is assigned if the format is correct, otherwise 0. In summary, the total reward for a response $o_i$ is calculated as: $r_i = r^i_{\text{acc}} + r^i_{\text{format}}$. To quantify the relative quality of all responses $\{r_1, \ldots, r_G\}$, GRPO normalizes these scores by subtracting the group mean and dividing by the standard deviation. Consequently, the advantage for each response can be formulated by:

$$A_i = \frac{r_i - \text{mean}(\{r_i\}_{i=1}^G)}{\text{std}(\{r_i\}_{i=1}^G)}, \tag{1}$$

where $A_i$ is the relative advantage of the $i$-th answer. Then the optimization objective further incorporates a regularization term to ensure the updated policy $\pi_\theta$ remains close to the original reference

policy $\pi_{\text{ref}}$. This is achieved by adding a KL-divergence term $D_{\text{KL}}(\cdot \| \cdot)$ to the loss function:

$$\mathcal{L}_{GRPO}(\theta) = -\mathbb{E}[q{\sim}P(Q), \{o_i\}_{i=1}^{G}\pi_{\theta\,old}(O|q)]\frac{1}{G}\sum_{i=1}^{G}$$

$$\left( min\left( \frac{\pi_\theta(o_i|q)}{\pi_{\theta\,old}(o_i|q)} * Adv_i, clip\left( \frac{\pi_\theta(o_i|q)}{\pi_{\theta\,old}(o_i|q)}, 1-\epsilon, 1+\epsilon \right) * Adv_i \right) - \beta\mathbb{D}_{KL}(\pi_\theta||\pi_{ref}) \right), \tag{2}$$

$$\mathbb{D}_{KL}(\pi_\theta||\pi_{ref}) = \frac{\pi_{ref}(o_i|q)}{\pi_\theta(o_i|q)} - log\frac{\pi_{ref}(o_i|q)}{\pi_\theta(o_i|q)} - 1, \tag{3}$$

where $\beta$ acts as a hyperparameter to trade-off between exploration and stability during optimization.

## 2.3 Physics-Grounded Accuracy Rewards in GRPO

In autonomous driving, traditional reward function designs often focus solely on trajectory position error, while neglecting the complex constraints in geometric, dynamical, and temporal dimensions. To address this issue, we propose a physics-grounded reward framework that integrates spatial alignment, vehicle dynamics, and temporal continuity to comprehensively guide the model in generating safe, feasible, and comfortable driving strategies. This multi-dimensional approach not only ensures geometric accuracy but also explicitly incorporates the physical limitations of real-world vehicles and the perceptual requirements for motion smoothness, creating a holistic optimization objective.

**Spatial Alignment: Balancing Global Maneuverability.** The foundation of any trajectory reward function lies in its ability to align predicted paths with target routes. We define a spatial accuracy term $r_{\text{pos}}$ as the mean squared Euclidean distance between predicted and ground-truth coordinates:

$$r_{\text{pos}} = \frac{1}{N}\sum_{i=1}^{N}\left( (x^i - x_{\text{gt}}^i)^2 + (y^i - y_{\text{gt}}^i)^2 \right), \tag{4}$$

where $N$ denotes the number of time steps, $x^i, y^i$ represent predicted coordinates at the $i$-th time step, while $x_{\text{gt}}^i, y_{\text{gt}}^i$ correspond to the ground-truth values. This formulation prioritizes global path adherence by penalizing deviations across all time steps, ensuring the vehicle remains on the intended route. However, focusing only on minimizing position error may produce physical-implausible results. For instance, strictly following the shortest path might bring about abrupt steering or acceleration, which not only violate vehicle kinematics but also compromise passenger comfort. To balance geometric precision with practical feasibility, we introduce additional constraints derived from vehicle dynamics.

**Vehicle Dynamics: Bridging Perception and Control.** Autonomous driving systems must respect the real-world physical limitations, which are governed by steering kinematics and longitudinal dynamics. Ignoring them may result in trajectories that are impossible to execute (e.g., requiring infinite torque for abrupt steering changes) or uncomfortable for passengers. To ensure kinematic feasibility, we penalize deviations in steering angles through the following term $r_{\text{ste}}$:

$$r_{\text{ste}} = \frac{1}{N}\sum_{j=1}^{N}\left( \theta^j - \theta_{\text{gt}}^i \right)^2, \tag{5}$$

where $\theta^j$ and $\theta_{gt}^j$ respectively denotes the predicted and corresponding ground-truth steering angle at $j$-th time step. Additionally, we address unphysical acceleration/braking patterns by introducing an additional velocity constraint term:

$$r_{\text{vel}} = \frac{1}{N}\sum_{k=1}^{N}\left( v^k - v_{\text{gt}}^k \right)^2, \tag{6}$$

where $v^k$ and $v_{gt}^j$ respectively represents the predicted and corresponding ground-truth velocity at the $k$-th time step. In summary, both $r_{\text{ste}}$ and $r_{\text{vel}}$ enforce compliance with vehicle-specific constraints, ensuring generated trajectories are both physically realizable and socially acceptable in

Table 1: Trajectory L2 errors and collision rates on the nuScenes dataset.

| Method | L2 Error (m) ↓ | | | | Collision Rate (%) ↓ | | | |
|---|---|---|---|---|---|---|---|---|
| | 1s | 2s | 3s | Avg. | 1s | 2s | 3s | Avg. |
| *Open-source Generalist Vision Language Models* | | | | | | | | |
| Llama-3.2-11B-Vision (Touvron et al., 2023) | 1.54 | 3.31 | 3.91 | 2.92 | - | - | - | - |
| DeepSeek-VL2-16B (Liu et al., 2025) | 0.66 | 1.68 | 2.92 | 1.75 | - | - | - | - |
| LLaMA-3.2-11B-Vision (Touvron et al., 2023) | 0.52 | 1.42 | 2.68 | 1.54 | - | - | - | - |
| Qwen-2.5-VL-3B (Bai et al., 2025) | 2.69 | 4.16 | 5.78 | 4.21 | 0.32 | 0.54 | 0.78 | 0.55 |
| Qwen-2.5-VL-7B (Bai et al., 2025) | 0.46 | 1.33 | 2.55 | 1.45 | 0.10 | 0.18 | 0.24 | 0.17 |
| *Training-based Driving Specialists (Existing Methods)* | | | | | | | | |
| UniAD (Hu et al., 2023) | 0.42 | 0.64 | 0.91 | 0.66 | 0.62 | 0.58 | 0.63 | 0.61 |
| VAD (Jiang et al., 2023) | 0.17 | 0.34 | 0.60 | 0.37 | 0.07 | 0.17 | 0.41 | 0.22 |
| BEV-Planner (Li et al., 2024) | 0.16 | 0.32 | 0.57 | 0.35 | 0.00 | 0.29 | 0.73 | 0.34 |
| Ego-MLP (Zhai et al., 2023) | 0.15 | 0.32 | 0.59 | 0.35 | - | - | - | - |
| *Ours and Key Competitors (Specialized Driving Models)* | | | | | | | | |
| DriveVLM (Tian et al., 2024) | 0.18 | 0.34 | 0.68 | 0.40 | 0.10 | 0.22 | 0.45 | 0.27 |
| OmniDrive (Wang et al., 2024) | 0.14 | 0.29 | 0.55 | 0.33 | 0.00 | 0.13 | 0.78 | 0.30 |
| DriveVLM-Dual (Tian et al., 2024) | 0.15 | 0.29 | 0.48 | 0.31 | 0.05 | 0.08 | 0.17 | 0.10 |
| EMMA (Hwang et al., 2024) | 0.14 | 0.29 | 0.54 | 0.32 | - | - | - | - |
| EMMA+ (Hwang et al., 2024) | 0.13 | 0.27 | 0.48 | 0.29 | - | - | - | - |
| Imprompt-VLA | 0.13 | 0.27 | 0.53 | 0.30 | - | - | - | - |
| **AutoDrive-R² 3B** | 0.35 | 0.49 | 0.62 | 0.49 | 0.13 | 0.44 | 0.87 | 0.48 |
| **AutoDrive-R² 7B** | 0.13 | 0.19 | 0.25 | 0.19 | 0.00 | 0.07 | 0.12 | 0.07 |

mixed traffic scenarios. These constraints explicitly bridge the gap between perception-driven planning and actuator-level control, ensuring that predicted trajectories align with the physical boundaries while maintaining ride quality.

**Temporal Smoothness: Ensuring Navigation Reliability** Temporal discontinuities in trajectory predictions fundamentally undermine the reliability of autonomous driving systems. When steering or acceleration commands exhibit sudden jumps between time steps, the predicted trajectories may lose coherence, which further compromises the system's ability to maintain stable, predictable motion patterns required for safe navigation. To address this, we introduce a temporal smoothness term $r_{\text{tem}}$ that penalizes rapid variations in consecutive control signals:

$$r_{\text{tem}} = \frac{1}{N} \sum_{j=1}^{N} \left( \theta^j - \theta^{j-1} \right)^2 + \frac{1}{N} \sum_{k=1}^{N} \left( v^k - v^{k-1} \right)^2. \tag{7}$$

Such design ensures temporal coherence of predicted trajectories. By explicitly constraining the rate of change in both steering and velocity, the reward function filters out unstable oscillations that could destabilize the vehicle's state estimation. This regularization effect strengthens the model's ability to generalize across diverse driving scenarios while maintaining safety margins during execution.

**Integrated Reward Function.** The final reward synthesizes all dimensions with learnable weights:

$$r_{\text{acc}} = \lambda_{\text{pos}} \cdot r_{\text{pos}} + \lambda_{\text{ste}} \cdot r_{\text{ste}} + \lambda_{\text{vel}} \cdot r_{\text{vel}} + \lambda_{\text{tem}} \cdot r_{\text{tem}}. \tag{8}$$

Here, $\lambda_{\text{pos}}, \lambda_{\text{ste}}, \lambda_{\text{vel}}, \lambda_{\text{tem}}$ are learnable coefficients that balance trade-offs between competing objectives. We set them all equal one in experiments. This holistic formulation ensures the model generates trajectories that are geometrically accurate, dynamically feasible, and temporally smooth, addressing the multifaceted challenges of autonomous driving.

## 3 EXPERIMENT

### 3.1 EXPERIMENTAL SETTINGS

**Datasets.** For training, we adopt nuScenesR$^2$-6K dataset, which contains 6k image-trajectory sample pairs, each includes a front-view image and a 3-second trajectory planning with 0.5-second intervals. The Qwen2.5-VL-7B model is fine-tuned on these samples for SFT to establish foundational perception capabilities before RL. For evaluation, our method is tested on nuScenes and Waymo datasets, both offering comprehensive autonomous driving data. The nuScenes dataset contains 1,000 urban driving scenes with six synchronized camera views to support planning tasks. Waymo dataset includes 4,021 driving segments, capturing eight camera views and ego-vehicle trajectories.

**Details.** We implement experiments on both Qwen2.5-VL-3B and Qwen2.5-VL-7B models. In both stages, the learning rate is set to 5e-7 with an accumulated total batch size of 8. The GRPO is configured with a maximum completion length of 4,096 tokens and samples 6 responses per input. For training, the reinforcement learning (RL) pipeline utilized the TRL framework, and was executed for a total of 750 training iterations and 1 epoch for 18 hours.

**Evaluation Metrics.** We evaluate performance using both accuracy and safety metrics. For accuracy, we adopt the L2 distance (in meters) between the predicted and ground truth trajectories at 1s, 2s, and 3s, along with the average error. To assess safety, we also report the Collision Rate (%), which measures the frequency of collisions in the predicted paths. For all models, we utilize the official checkpoints and conduct evaluations under the same evaluation codes to ensure fairness. Note that the best and second-best results are highlighted in all tables.

### 3.2 EVALUATION RESULTS

**Results on nuScenes Datasets** Table 1 compares the prediction errors among our method and existing approaches on the nuScenes dataset. Notably, our approach consistently achieves the best performance across all time intervals, surpassing current leading methods such as EMMA+, which are trained on substantially larger internal datasets with 103k scenarios. In contrast, our training data consists of only 6k curated CoT samples for stage 1 and another 6k for stage 2, approximately 11.65% the size of EMMA+'s dataset. Furthermore, our model demonstrates significant improvements over Qwen2-VL-7B, reducing L2 errors by 86.9%, despite having less parameter.

**Zero-shot performance on Waymo Datasets** Moreover, Tab. 2 demonstrates the robust zero-shot capabilities of our model. Specifically, our method respectively reduces L2 errors by 33.3% and 90.7% compared to the latest EMMA+ method and Qwen2-VL-72B baseline models. Overall, our model consistently delivers precise trajectory predictions across multiple datasets, establishing its state-of-the-art performance and generalization capability.

**Model Size** In Tab. 1 and Tab. 2, we compare 3B and 7B variants of Qwen2.5-VL within our two-stage training framework to analyze impact of different model size. While the 7B model achieves superior performance with an average L2 error of

Table 2: Trajectory L2 errors on Waymo.

| Method | L2 Error (m) ↓ | | | |
|---|---|---|---|---|
| | 1s | 2s | 3s | Avg. |
| *Generalist VLMs + Specialized Driving Models* | | | | |
| Qwen-2.5-VL-3B | 2.98 | 5.05 | 7.38 | 5.14 |
| Qwen-2.5-VL-7B | 1.66 | 1.82 | 2.92 | 2.13 |
| DriveVLM | 0.17 | 0.34 | 0.75 | 0.42 |
| EMMA | 0.12 | 0.28 | 0.61 | 0.34 |
| EMMA+ | 0.11 | 0.25 | 0.53 | 0.30 |
| **AutoDrive-R² 3B** | 0.23 | 0.36 | 0.51 | 0.37 |
| **AutoDrive-R² 7B** | 0.11 | 0.19 | 0.29 | 0.20 |

0.19m, the 3B version demonstrates a notable improvement than its baseline. The disparity highlights that larger models inherently capture more complex patterns, but the two-stage framework (SFT + GRPO) effectively compensates for the 3B model's limited capacity by enforcing strict trajectory constraints and contextual logic synthesis.

**Closed-loop Experiment** To evaluate our model's practical decision-making, we conducted closed-loop experiments on the NAVSIM benchmark. As detailed in Table 3, our method sets a new state-of-the-art by outperforming all prior approaches across every metric. Specifically, our model achieves top scores in safety, with a Navigation Completion of 98.3 and Time-to-Collision of 95.6, while also excelling in driving quality by leading in Driving Agent Comfort (94.4) and Ego-Progress (81.6).

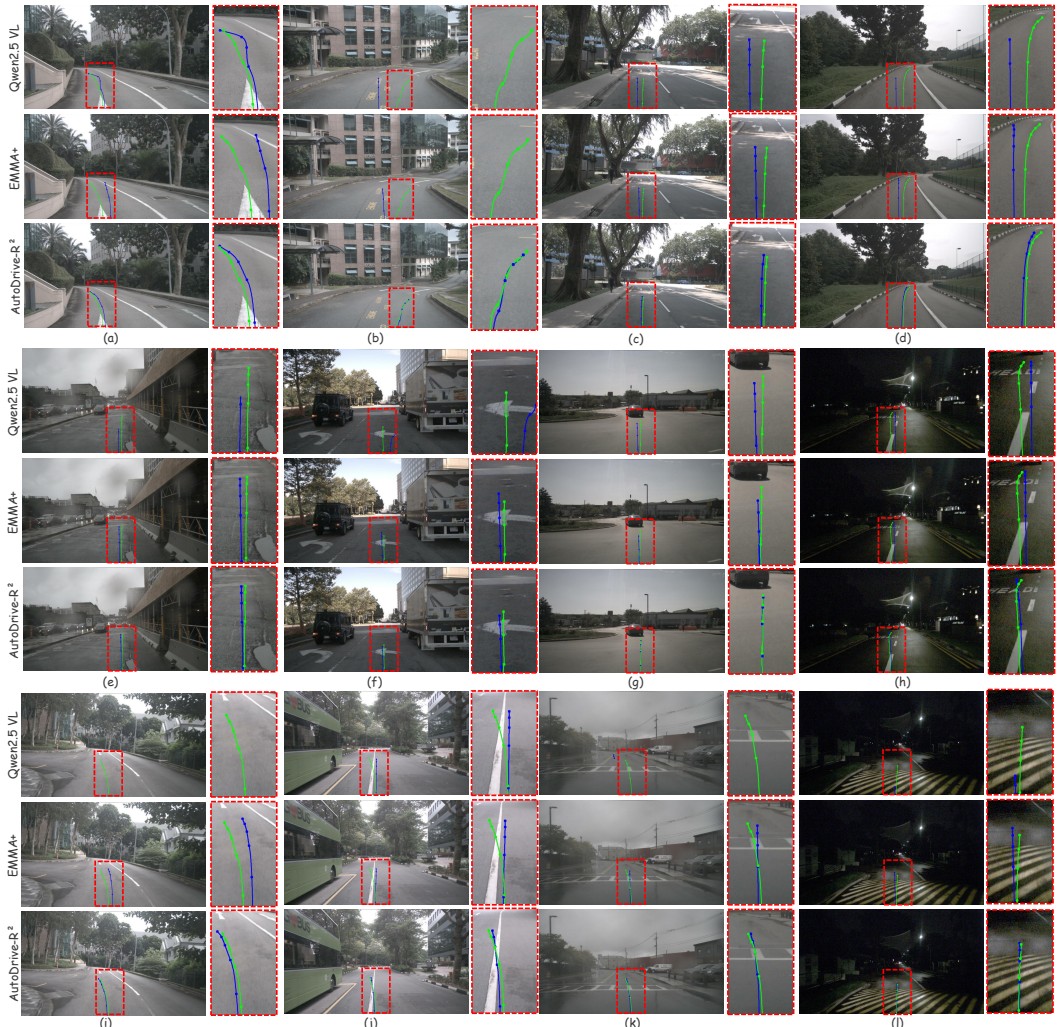

Figure 2: Qualitative comparison of trajectory planning performance across Qwen2.5-VL-7B, EMMA+, and our AutoDrive-R² on the nuScenes dataset. Note that blue lines denote predicted trajectories while green lines represent ground truth trajectories.

The most significant advantage is observed in the Planner-Diverse Motion Score (PDMS), where our score of 90.3 surpasses the next-best competitor by over 6 points, showcasing superior human-like planning. These comprehensive results confirm our model's ability to translate its predictive accuracy into robust, safe, and efficient real-world driving actions.

Table 3: Performance on the Closed-loop NAVSIM.

| Methods | NC↑ | DAC↑ | TTC↑ | Comfort↑ | EP↑ | PDMS↑ |
|---|---|---|---|---|---|---|
| TransFuser (Chitta et al., 2022) | 97.7 | 92.8 | 92.8 | 100 | 79.2 | 84.1 |
| UniAD (Hu et al., 2023) | 97.8 | 91.9 | 92.9 | 100 | 78.8 | 83.4 |
| Para-Drive (Weng et al., 2024b) | 97.9 | 92.4 | 93.0 | 99.8 | 79.3 | 84.0 |
| **AutoDrive-R² 7B** | 98.5 | 95.9 | 95.4 | 100 | 82.7 | 89.1 |

**Visualization** In Fig. 2, we present a comparative analysis of our method against other approaches in the nuScenes dataset. Notably, Qwen2.5-VL-7B fails to generate accurate predictions in specific scenarios (e.g., (b) and (d)), whereas EMMA+ exhibits significant trajectory deviation. In contrast, our method consistently achieves more reliable and physically feasible trajectory planning under varying illumination environments and complex motion patterns.

## 3.3 Ablation Studies

**Training Stages** Drawing inspiration from DeepSeek-R1-Zero, we first attempt to train the model solely through RL. As shown in Tab. 4, the model purely trained on RL (7B + *RL*) underperforms that of SFT (7B + *SFT*) by 22.2% in average L2 error. We attribute this to model's inability to establish structured reasoning chains, as RL struggles to explore the high-dimensional reasoning space required for multi-step calculations and contextual logic synthesis. This observation validates the necessity of our two-stage training.

**Supervised Fine-tuning (*SFT*)** In the first stage, the baseline Qwen2.5-VL-7B (7B) achieves an average L2 error of 1.45m, whereas the SFT model (7B + *SFT*) trained on the nuScenesR²-6K dataset reduces this to 0.27m, demonstrating an 81.4% improvement. This significant enhancement highlights the effectiveness of adopting SFT training in establishing foundational reasoning capabilities.

Table 4: Ablation studies of trajectory L2 errors on nuScenes dataset for validation.

| Method | L2 Error (m) ↓ | | | |
|---|---|---|---|---|
| | 1s | 2s | 3s | Avg. |
| Qwen2.5-VL-7B | 0.46 | 1.33 | 2.55 | 1.45 |
| Qwen2.5-VL-7B + *SFT* | 0.17 | 0.27 | 0.36 | 0.27 |
| Qwen2.5-VL-7B + *RL* | 0.21 | 0.33 | 0.44 | 0.33 |
| *SFT* : w/o. Four. | 0.19 | 0.25 | 0.32 | 0.25 |
| *SFT* : w/o. Self. | 0.17 | 0.23 | 0.29 | 0.23 |
| *RL* : w/o. $r_{pos}$ | 0.32 | 0.53 | 0.72 | 0.53 |
| *RL* : w/o. $r_{ste}$ | 0.14 | 0.20 | 0.27 | 0.21 |
| *RL* : w/o. $r_{vel}$ | 0.15 | 0.21 | 0.29 | 0.22 |
| *RL* : w/o. $r_{tem}$ | 0.15 | 0.23 | 0.34 | 0.24 |
| **AutoDrive-R² 7B** | 0.13 | 0.19 | 0.25 | 0.19 |

establishing foundational reasoning capabilities. Moreover, removing four-step reasoning structure (w/o. Four.) increases the error to 0.25m, indicating a 31.5% degradation compared to AutoDrive-R². Similarly, eliminating self-reflection (w/o. Self.) results in 0.23m error, representing a 21.1% decline relative to AutoDrive-R². This emphasizes the interdependence of both four-step logical chain and self-reflection mechanism in constructing high-quality CoT dataset.

**Reinforcement Learning (*RL*)** In the second stage, we evaluate the contribution of individual reward components within the physics-grounded framework of AutoDrive-R². Specifically, spatial alignment is critical for maintaining global geometric path accuracy, as its removal (w/o. $r_{pos}$) increases the error to 0.53m, much higher than the full model. Moreover, steering angle regulation ensures kinematic feasibility by penalizing abrupt changes in steering adjustments, and its absence (w/o. $r_{ste}$) leads to a 10.5% degradation (0.21m). Additionally, velocity consistency constraints ensure adherence to target speed profiles by penalizing deviations in predicted velocity from ground-truth values, and their exclusion (w/o. $r_{vel}$) raises the error to 0.22m. Finally, temporal smoothness penalties suppress unstable control patterns by penalizing abrupt changes in steering and velocity across time steps, and their removal (w/o. $r_{tem}$) results in a 26.3% increase in error (0.24m). By integrating all four components into our physics-grounded reward, AutoDrive-R² achieves an optimal 0.19m L2 error, confirming the necessity of each element in achieving reliable trajectory planning.

**Key Hyperparameters** In Tab. 5, We conduct a series of ablation studies to analyze the impact of key hyperparameters. First, for the reward weights ($\lambda$), we find that uniform weights ($\lambda = (1, 1, 1, 1)$) achieve a lower average L2 error (0.19 m) compared to decaying weights (0.22 m). This indicates that all reward components are equally important for the learning process, leading us to adopt the uniform setting. Next, we analyze the KL-divergence coefficient $\beta$. A value of $\beta = 0.04$ achieves the best performance with a 0.19 m error. A smaller value ($\beta = 0.02$) results in a higher error (0.21 m), while a larger value ($\beta = 0.06$) also slightly degrades performance (0.20 m), likely by overly constraining the optimization. Finally, for the number of generations ($G$), we observe performance improving as $G$ increases from 2 (0.23 m) to 6 (0.19 m). However, increasing $G$ further to 8 yields no additional benefit. Therefore, we set $G = 6$ to balance performance and computation.

Table 5: Ablations on key hyperparameters.

| Setting | L2 Error (m) ↓ | | | |
|---|---|---|---|---|
| | 1s | 2s | 3s | Avg. |
| *Reward Weights ($\lambda$)* | | | | |
| $\lambda : (0.4, 0.3, 0.2, 0.1)$ | 0.15 | 0.22 | 0.29 | 0.22 |
| $\lambda : (1, 1, 1, 1)$ **(Ours)** | 0.13 | 0.19 | 0.25 | 0.19 |
| *KL-divergence Beta ($\beta$)* | | | | |
| $\beta = 0.02$ | 0.14 | 0.21 | 0.27 | 0.21 |
| $\beta = 0.04$ **(Ours)** | 0.13 | 0.19 | 0.25 | 0.19 |
| $\beta = 0.06$ | 0.13 | 0.20 | 0.26 | 0.20 |
| *Number of Generations ($G$)* | | | | |
| $G = 2$ | 0.16 | 0.23 | 0.31 | 0.23 |
| $G = 4$ | 0.14 | 0.20 | 0.26 | 0.20 |
| $G = 6$ **(Ours)** | 0.13 | 0.19 | 0.25 | 0.19 |
| $G = 8$ | 0.13 | 0.19 | 0.25 | 0.19 |

## 4 RELATED WORK (EXTENDED VER. IN APPX. B)

**Autonomous Driving** The evolution of autonomous driving systems has transitioned from modular architectures to end-to-end learning paradigms. Early works like UniAD (Hu et al., 2023) pioneered the integration of sub-tasks into a unified framework. Concurrently, other methods addressed core challenges in prediction and efficiency; for example, Flash (Antonello et al., 2022) focused on creating fast models for real-time application, while DiPA (Knittel et al., 2023) tackled the complex, multi-modal nature of agent interactions. These works highlighted the need for a holistic evaluation beyond simple L2 error, inspiring our new analyses on collision rate and closed-loop performance. Subsequent methods like Para-Drive (Weng et al., 2024a) and BEV-Planner (Li et al., 2024) continued to refine these integrated approaches.

The advent of vision-language models (VLMs) has further transformed the field. DriveVLM (Tian et al., 2024) and DriveMLM (Wang et al., 2023) enabled systems to incorporate linguistic reasoning for decision-making and rationale generation. Alongside performance, explainability has become a critical concern. A systematic review in (Kuznietsov et al., 2024) provides a comprehensive overview of techniques for building trustworthy systems, while works like Interpretable Goal-based Prediction and Planning (Albrecht et al., 2021) explored generating explanations by making the model's goals explicit. In contrast to these post-hoc or explicit-goal methods, our AutoDrive-R² introduces a novel form of *intrinsic* explainability, representing a significant step towards inherently trustworthy AI for autonomous driving.

**General VLMs** The development of general vision-language models (VLMs) has been driven by the success of large language models (LLMs) in understanding textual data. CLIP (Radford et al., 2021) established a foundational approach by aligning image and text features through contrastive learning, enabling zero-shot generalization. Building on this, BLIP (Li et al., 2022) and its successor BLIP-2 (Li et al., 2023) refined multimodal alignment using contrastive and matching losses to improve contextual grounding. More recent models, such as LLaVA (Liu et al., 2023) and Qwen2.5VL (Bai et al., 2025), have integrated robust LLMs with vision encoders to enhance representation capabilities. The OmniGen2 (Wu et al., 2025) framework further advanced this by introducing dual decoding pathways for text and image generation, while DeepSeek-V3 (Liu et al., 2025) demonstrated efficient inference through a Mixture-of-Experts (MoE) architecture with auxiliary-loss-free load balancing. These advancements highlight the growing synergy between vision and language modalities in multimodal learning.

**Reinforcement Learning for Post-Training** Reinforcement learning (RL) has emerged as a critical tool for refining model capabilities post-training. Techniques like Proximal Policy Optimization (PPO) (Schulman et al., 2017) have been instrumental in fine-tuning models for complex tasks, as seen in the optimization of GPT (Achiam et al., 2024). Direct Preference Optimization (DPO) (Rafailov et al., 2023) introduces a sampling-free parameterization for reward models, streamlining the fine-tuning process. Reward Fine-Tuning (RFT) (Yuan et al., 2023) has shown particular efficacy in mathematical reasoning, while Guided Reward Policy Optimization (GRPO) (Shao et al., 2024b) enables robust reasoning improvements without external toolkits. The DeepSeek-R1 (Guo et al., 2025) model exemplifies the application of GRPO, achieving state-of-the-art results through physics-informed reward design. These methodologies underscore the importance of RL in aligning model outputs with real-world constraints and user preferences.

## 5 CONCLUSION

In this work, we propose AutoDrive-R², a novel VLA framework designed for reasoning-guided trajectory planning in autonomous driving. AutoDrive-R² effectively balances semantic understanding with real-world constraints through a two-stage training framework: (1) a SFT stage adopting the nuScenesR²-6K dataset, which employs a four-step CoT process to cultivate structured reasoning and self-reflection for validation, and (2) a RL stage leveraging GRPO training to refine trajectory planning under physics-grounded rewards. Experiments validate the effectiveness of AutoDrive-R², achieving SOTA performance on nuScenes (34.5% reduction vs. EMMA+) and Waymo (90.7% reduction vs. Qwen2.5-VL-7B), demonstrating strong zero-shot generalization. Future efforts will focus on multi-agent coordination and real-time sensor fusion integration to further improve adaptability in complicated environments.

## ETHICS STATEMENT

Our research builds upon publicly available autonomous driving datasets, including nuScenes and Waymo, to construct the nuScenesR²-6K dataset. The manual annotation process focuses exclusively on labeling contextual reasoning chains and trajectory planning scenarios, ensuring no personally identifiable information (PII) is collected or inferred. The dataset is fully anonymized, emphasizing high-level environmental and traffic dynamics while safeguarding individual privacy. We advocate for the responsible application of AutoDrive-R², urging stakeholders to prioritize safety, fairness, and transparency in deployment. Specifically, we caution against misuse in surveillance or discriminatory decision-making, particularly in scenarios where autonomous systems could disproportionately impact vulnerable populations. All experiments adhere to rigorous academic standards, with a commitment to open science: we publicly share our dataset, code, and training protocols to enable independent verification, ethical scrutiny, and collaborative refinement. By fostering transparency and community engagement, we aim to ensure that advancements in autonomous driving align with societal values and regulatory frameworks.

## REPRODUCIBILITY STATEMENT

To ensure the full reproducibility of our findings, we have provided detailed implementation and methodological descriptions throughout the paper. The construction of the nuScenesR²-6K dataset, including its 6,000 manually annotated image-trajectory pairs and statistical properties, is described in Sec. 2.1 and Appendix D. The AutoDrive-R² framework, with its four-step reasoning process (Observation → Calculation → Logic → Reflection) and self-reflection validation mechanism, is thoroughly outlined in Sec. 2.2, where we also provide the structured input prompts used to generate chain-of-thought (CoT) data. Key components of the Group Relative Policy Optimization (GRPO) training algorithm, including the physics-grounded reward function, spatial alignment, vehicle dynamics, and temporal smoothness criteria, are detailed in Sec. 2.3 and Appendix C. To further facilitate replication, we will publicly release the nuScenesR²-6K dataset, pre-trained model checkpoints, and training code upon acceptance, adhering to open scientific principles. Additionally, all hyperparameters, evaluation protocols, and ablation study configurations are explicitly documented in Appendix A, ensuring transparent and rigorous experimental validation.

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

# APPENDIX

In this appendix, we provide more details, related work, tool library, and discussions for a comprehensive evaluation and understanding of our method. Detailed contents are as follows:

## A EXPERIMENT

### A.1 MORE HYPERPARAMETER CONFIGURATIONS

Our method is implemented on a machine with an Intel(R) Xeon(R) Platinum 8480+ and eight 8 NVIDIA H20 GPUs with 90G memory. The training process ran for 750 epochs without freezing the vision transformer (ViT) backbone, and the number of generation is set to 6 in GRPO algorithm.

### A.2 MORE VISUALIZATION RESULTS

**Fig. 3** provides additional visualization results in trajectory planning tasks between our AutoDrive-R² and other methods on the nuScenes dataset. Notably, our method consistently outperforms other approaches in predicting both reliable and physically-feasible trajectories, demonstrating the state-of-the-art performance of our proposed method.

To further illustrate the advantages of our structured reasoning process, we present two representative comparisons in **Fig. 6 and Fig. 7**. These visualizations explicitly contrast the four-stage CoT reasoning (AutoDrive-R²) with the single-step reasoning of Qwen2.5-VL-7B. As can be seen, the Qwen2.5-VL model predicts a trajectory that deviates from the lane marking due to its simplified reasoning approach. The model's single-stage analysis fails to account for the vehicle's kinematic constraints and results in an unrealistic leftward drift. In contrast, AutoDrive-R²'s four-stage process systematically validates its predictions. These examples demonstrate how our structured CoT framework enables systematic error detection and correction, resulting in trajectories that are both geometrically accurate and physically feasible. The integration of physics-grounded rewards in the GRPO stage further ensures these corrections align with real-world driving constraints.

### A.3 "AHA MOMENT"

A compelling insight observed during the development of AutoDrive-R² is the emergence of a **"reasoning self-correction moment"**, where the model systematically identifies and resolves contradictions in its initial trajectory planning.

This reasoning self-correction moment demonstrates AutoDrive-R²'s ability to re-examine its own assumptions and refine predictions through structured self-validation. As shown in **Fig. 5**, similar

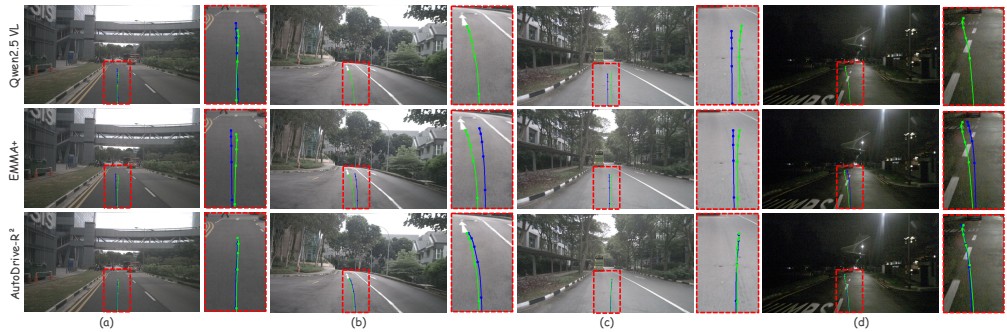

Figure 3: Qualitative comparison of trajectory planning performance across Qwen2.5-VL-7B, EMMA+, and our AutoDrive-R² on the nuScenes dataset. Note that blue lines denote predicted trajectories while green lines represent ground truth trajectories.

**"task":** "You are an autonomous driving agent. You have access to a front view camera image of a vehicle <image>. Your task is to do your best to predict future waypoints for the vehicle over the next 3 timesteps, given the vehicle's intent inferred from the images."
**"information":** "Provided are the previous ego vehicle status recorded over the last 3.0 seconds (at 0.5-second intervals). This includes the x and y coordinates of the ego vehicle. Positive x means forward direction while positive y means leftwards. The data is presented in the format [x, y]:.(t-3.0s) [-12.37, -0.04], Acceleration: X -0.06, Y -0.12 m/s^2, Velocity: 5.1 m/s, Steering angle: -0.19 (positive: left turn, negative: right turn), (t-2.5s) [-9.93, 0.02], Acceleration: X -0.13, Y -0.29 m/s^2, Velocity: 4.73 m/s, Steering angle: -0.16 (positive: left turn, negative: right turn), (t-2.0s) [-7.68, 0.04], Acceleration: X 0.11, Y -0.06 m/s^2, Velocity: 4.41 m/s, Steering angle: -0.13 (positive: left turn, negative: right turn), (t-1.5s) [-5.57, 0.03], Acceleration: X 0.08, Y 0.03 m/s^2, Velocity: 4.26 m/s, Steering angle: -0.05 (positive: left turn, negative: right turn), (t-1.0s) [-3.46, 0.01], Acceleration: X -0.37, Y -0.02 m/s^2, Velocity: 3.94 m/s, Steering angle: -0.05 (positive: left turn, negative: right turn), (t-0.5s) [-1.67, 0.01], Acceleration: X -0.4, Y -0.33 m/s^2, Velocity: 3.6 m/s, Steering angle: 0.05 (positive: left turn, negative: right turn), (t-0.0s) [0.0, 0.0], Acceleration: X -0.92, Y -0.12 m/s^2, Velocity: 3.03 m/s, Steering angle: 0.06 (positive: left turn, negative: right turn)",

**Input Image:**

AutoDrive-R²

**AutoDrive-R²:** <think>1. Visual Analysis** \"Image analysis results: - Vehicle's intended direction: The vehicle appears to be moving straight ahead, as there are no visible signs of a left or right turn. The steering angle is minimal and fluctuates slightly, suggesting a straight path. * No pedestrians or other vehicles are visible in the immediate vicinity. - Traffic signal: There is no visible traffic signal in the image. The barrier and the road markings suggest a controlled access point, but no traffic light is present.\"### **2. Motion Modeling** \"Using historical data in **Past Vehicle Status** with 7 time points: t₁: [x=-12.37, y=-0.04], v=5.1m/s, a=(-0.06, -0.12)m/s² t₂: [x=-9.93, y=0.02], v=4.73m/s, a=(-0.13, -0.29)m/s² ... t₇: [x=0.0, y=0.0], v=3.03m/s, a=(-0.92, -0.12)m/s² Calculations: - Average acceleration: a_avg_x = (-0.06 - 0.13 + 0.11 + 0.08 - 0.37 - 0.4 - 0.92) / 7 ≈ -0.25m/s² a_avg_y = (-0.12 - 0.29 - 0.06 + 0.03 - 0.02 - 0.33 - 0.12) / 7 ≈ -0.15m/s² - Velocity prediction: v_x = 3.03 + (-0.25) × 0.5 ≈ 2.86m/s v_y = 0 + (-0.15) × 0.5 ≈ -0.075m/s - Position prediction: x(t+1) = 0 + 2.86 × 0.5 + 0.5 × (-0.25) × (0.5)² ≈ 1.39m y(t+1) = 0 + (-0.075) × 0.5 + 0.5 × (-0.15) × (0.5)² ≈ -0.02m - Lateral offset: Δy = v × tan(θ) ≈ 3.03 × tan(0.06) ≈ 0.018m\"### **3. Logical Deductions** \"Safety check: - If following this trajectory, will the vehicle: - Run a red light? → no, as there is no traffic light visible. * Collide with car ahead? → no, as the road is clear ahead of the vehicle. * Hit pedestrian crossing? → no, as no pedestrians are visible. - Conclusion: **Continue moving forward at constant speed** \"### **4. Self-Reflection Validation** \"Validation: - Assumption check: * Predicted position (x=1.39, y=-0.02) requires an average speed of approximately 2.86m/s. * Is this speed achievable with vehicle's acceleration history? → yes, as the vehicle has been decelerating and the predicted speed is within the range of previous velocities. - Adjustment: * If not feasible → Modify trajectory by reducing speed further or increasing stopping distance. **Let me think, the vehicle is decelerating and the predicted speed is within the range of previous velocities. Wait, I just realized that there's a physical barrier** directly ahead blocking the entire lane; the vehicle cannot possibly navigate around it by continuing forward as previously assumed. **Oh, I see, the entire premise of moving forward is flawed because the road is impassable.** Let's break it down: the vehicle must come to a complete stop before reaching the barrier to avoid a collision. The predicted waypoints are not feasible or safe in this context. The correct action is to halt. Based on the current velocity of 3.03 m/s and the maximum feasible deceleration from the historical data (around -0.92 m/s²), the vehicle can achieve a full stop in approximately 1.5 seconds. The predicted waypoints should therefore reflect this faster, controlled deceleration, bringing the vehicle to rest closer to its starting position while ensuring safety and comfort. <answer>[0.45, -0.01], [0.65, -0.02], [0.70, -0.03], [0.70, -0.03], [0.70, -0.03], [0.70, -0.03]</answer>

Figure 4: "Aha Moment" of our AutoDrive-R² on the nuScenes dataset.

**"task":** "You are an autonomous driving agent. You have access to a front view camera image of a vehicle <image>. Your task is to do your best to predict future waypoints for the vehicle over the next 3 timesteps, given the vehicle's intent inferred from the images."
**"information":** "Provided are the previous ego vehicle status recorded over the last 3.0 seconds (at 0.5-second intervals). This includes the x and y coordinates of the ego vehicle. Positive x means forward direction while positive y means leftwards. The data is presented in the format [x, y]:.(t-3.0s) [-14.86, -2.52], Acceleration: X 0.25, Y 0.44 m/s^2, Velocity: 3.82 m/s, Steering angle: -1.0 (positive: left turn, negative: right turn), (t-2.5s) [-12.81, -1.87], Acceleration: X 0.17, Y 0.34 m/s^2, Velocity: 4.35 m/s, Steering angle: -0.75 (positive: left turn, negative: right turn), (t-2.0s) [-10.5, -1.24], Acceleration: X -0.12, Y 0.35 m/s^2, Velocity: 4.84 m/s, Steering angle: -0.59 (positive: left turn, negative: right turn), (t-1.5s) [-7.93, -0.67], Acceleration: X -0.29, Y 0.15 m/s^2, Velocity: 5.21 m/s, Steering angle: -0.7 (positive: left turn, negative: right turn), (t-1.0s) [-5.15, -0.21], Acceleration: X -0.59, Y -0.2 m/s^2, Velocity: 5.51 m/s, Steering angle: -1.04 (positive: left turn, negative: right turn), (t-0.5s) [-2.3, 0.01], Acceleration: X -1.07, Y -0.4 m/s^2, Velocity: 5.6 m/s, Steering angle: -1.13 (positive: left turn, negative: right turn), (t-0.0s) [0.0, 0.0], Acceleration: X -1.16, Y -0.59 m/s^2, Velocity: 5.64 m/s, Steering angle: -1.56 (positive: left turn, negative: right turn)",

**Input Image:**

AutoDrive-R²

**AutoDrive-R²:** <think> 1. Visual Analysis** \"Image analysis results: - Vehicle's intended direction: The vehicle is approaching a sharp rightward curve, with the immediate forward path blocked by a steep, grassy embankment and a wire-mesh fence on the left—no viable straight-through route exists. The road surface ahead appears clear, but the terrain forces a stop or turn. - Obstacles ahead: * A large, impassable earthen slope occupies the left and central forward field, acting as a physical barrier to straight-line travel. * Two orange traffic cones and a white sign are visible near the curve, indicating a controlled area requiring caution. * No pedestrians or other vehicles are present in the drivable corridor. - Traffic signal: None visible; this appears to be a low-traffic access road with passive control only.\" ### **2. Motion Modeling** \"Using historical data in **Past Vehicle Status** with N = 7 time points t₁: [x=-14.86, y=-2.52], v=3.82m/s, a=(0.25, 0.44)m/s² t₂: [x=-12.81, y=-1.87], v=4.35m/s, a=(0.17, 0.34)m/s² ... t₇: [x=0.0, y=0.0], v=5.64m/s, a=(-1.16, -0.59)m/s² Calculations: - Average acceleration: a_avg_x = (-1.16 -1.07 -0.59 -0.29 -0.12 + 0.17 + 0.25) / 7 ≈ -0.03m/s² a_avg_y = (-0.59 -0.40 -0.20 + 0.15 + 0.35 + 0.34 + 0.44) / 7 ≈ -0.03m/s² - Velocity prediction for deceleration: v_x = 5.64 + (-0.33) × 0.5 ≈ 5.52m/s v_y = 0 + (-0.03) × 0.5 ≈ -0.015m/s - Position prediction assuming continued deceleration: x(t+1) = 0 + 5.52 × 0.5 + 0.5 × (-0.33) × (0.5)² ≈ 2.71m y(t+1) = 0 + (-0.015) × 0.5 + 0.5 × (-0.03) × (0.5)² ≈ -0.04m - Lateral offset: Δy = v × tan(θ) = 5.64 × tan(-1.56°) ≈ -0.15m\" ### **3. Logical Deductions** \"Safety check: - If following this trajectory, will the vehicle: * Run a red light? → no (no traffic signal detected) * Collide with car ahead? → no (no cars detected) * Hit pedestrian crossing? → no (no pedestrians detected) - Conclusion: **Decelerate gradually to a full stop within 3 seconds to reassess environment**\" ### **4. Self-Reflection Validation** \"Validation: - Assumption check: The initial plan to stop assumes no alternative path, which aligns with the obstructed forward view. **Let me think, the vehicle's current velocity (5.64m/s) and negative acceleration suggest it's already slowing, but stopping within 3s requires sustained deceleration of ~-1.0m/s², feasible given history. Wait, I'm re-examining the right edge of the frame and notice the continuous white lane line curves smoothly rightward beyond 5m, not ending abruptly. Oh, I see, the embankment isn't a dead end; it's roadside topography, and the road itself continues along the curve.** The cones and sign mark a narrowing, not closure. Let's break it down: continuing to brake to a full stop is safe but suboptimal; initiating a gentle right turn at reduced speed (≈3.0m/s) while tracking the lane centerline is both physically possible and consistent with infrastructure cues. Self-validation confirms that turning right avoids unnecessary stopping, maintains momentum, and aligns with the road's design intent, yielding a revised waypoint sequence that follows the curvature.\" ### **Final Prediction** Based on corrected trajectory reasoning incorporating smooth rightward steering input (target lateral displacement -0.25m at 2.7m forward), the predicted future waypoints for the vehicle over the next 3 seconds are: <answer>[2.7, -0.25], [5.3, -0.90], [7.64, -1.95], [9.46, -2.88], [10.91, -3.72], [12.01, -4.35]</answer>

Figure 5: "Aha Moment" of our AutoDrive-R² on the nuScenes dataset.

behavior is observed during trajectory planning tasks: the model exhibits an emergent capacity to detect inconsistencies in its motion modeling and resolve them via iterative recalculations.

> *Let me think, the vehicle's current velocity (5.64m/s) and negative acceleration suggest it's already slowing, but stopping within 3s requires sustained deceleration of 1.0m/s², feasible given history.*
> *Wait, I'm re-examining the right edge of the frame and notice the continuous white lane line curves smoothly rightward beyond 5m, not ending abruptly.*
> *Oh, I see, the embankment isn't a dead end; it's roadside topography, and the road itself continues along the curve. The cones and sign mark a narrowing, not closure.*

This autonomous reasoning process, characterized by four-stage self-reflection, is a hallmark of AutoDrive-R²'s training pipeline. The integration of physics-grounded rewards in the GRPO stage ensures that even minor corrections (e.g., adjusting lateral offsets or deceleration rates) are validated against real-world constraints.

## B  RELATED WORK

### B.1  AUTONOMOUS DRIVING

In recent years, autonomous driving has evolved from traditional modular pipelines—comprising detection, online mapping, prediction, and planning—toward end-to-end learning-based approaches (Hu et al., 2023; Jiang et al., 2023; Sun et al., 2024). UniAD (Hu et al., 2023) was the first to integrate all sub-tasks into a cascaded model, achieving significant improvements over traditional modular approaches. Some methods (Jiang et al., 2023; Ye et al., 2023; Chen et al., 2024) extract bird's-eye view features and predict planning trajectories via multiple stages of interaction modeling. Para-Drive (Weng et al., 2024a) explores the design space of modular end-to-end autonomous driving stacks to significantly enhance both accuracy and inference speed. Additionally, Ego-MLP and BEV-Planner (Zhai et al., 2023; Li et al., 2024) investigate the role of ego-vehicle state and empirically validate its influence through experimental results.

With the emergence of vision-language models (VLMs), researchers have increasingly integrated both large language models (LLMs) and VLMs into autonomous driving to enhance overall system performance. Several approaches (Xu et al., 2024; Shao et al., 2024a) incorporate pretrained LLMs to generate driving actions along with interpretable textual explanations. Furthermore, DriveVLM (Tian et al., 2024) incorporates specialized reasoning modules to improve situational understanding, while DriveMM (Huang et al., 2024) processes multi-view video and image inputs to enhance generalization in vehicle control. DriveMLM (Wang et al., 2023) introduces a behavior planning module to generate optimal driving decisions with accompanying rationales.

Moreover, the recent success of vision-language-action (VLA) models in robotics offers a new perspective for autonomous driving. DriveMoE (Yang et al., 2025) builds on the embodied AI framework $\pi 0$ (Black et al., 2024) and introduces Action-MoE by training routing networks to activate expert modules for diverse driving behaviors. Furthermore, OpenDriveVLA (Zhou et al., 2025a) proposes an agent-environment-ego interaction model for precise trajectory planning. AutoVLA (Zhou et al., 2025b) directly predicts semantic reasoning and trajectory plans from visual inputs and language prompts.

### B.2  GENERAL VLMS

In recent years, the success of large language models (LLMs) (Yenduri et al., 2023; Brown et al., 2020; Touvron et al., 2023) has motivated researchers to extend them into vision-language models (VLMs) (Radford et al., 2021; Zhu et al., 2023), which integrate textual and visual data for richer multimodal representation. CLIP (Radford et al., 2021), a pioneering work, combines image and text features using an image encoder and a text encoder to predict correct pairings of image-text examples via a zero-shot learning strategy. Similarly, BLIP (Li et al., 2022) and BLIP-2 (Li et al., 2023) are trained using an image-text contrastive (ITC) loss to align vision and language representations, along with an image-text matching (ITM) loss to distinguish between positive and negative image-text pairs, thereby enhancing visual representation grounded in textual context. Inspired

by these methods, many VLMs—such as LLaVA (Liu et al., 2023) and Qwen2.5VL (Bai et al., 2025)—further enhance the robustness and representation capabilities of pretrained vision encoders by integrating a large language model as the text encoder (e.g., LLaMA (Touvron et al., 2023)). OmniGen2 (Wu et al., 2025) represents another notable VLM, employing two distinct decoding pathways for text and image modalities with unshared parameters and a decoupled image tokenizer. Notably, DeepSeek-V3 (Liu et al., 2025) introduces a robust Mixture-of-Experts (MoE) language model that employs an auxiliary-loss-free strategy for load balancing, achieving both efficient and cost-effective inference.

### B.3 REINFORCEMENT LEARNING FOR POST-TRAINING

Reinforcement learning (RL) has been widely adopted in large language models, and researchers have found that reinforcement learning from human feedback (RLHF) (Lambert, 2025) can significantly enhance their reasoning capabilities. Among these methods, Proximal Policy Optimization (PPO) (Schulman et al., 2017) was initially used in simulated robotic locomotion and Atari game environments, and later employed by OpenAI to fine-tune GPT (Achiam et al., 2024), resulting in substantial improvements in text generation tasks. Unlike conventional RLHF methods, direct Preference Optimization (DPO) introduces a new reward model parameterization that eliminates the need for sampling during fine-tuning (Rafailov et al., 2023). Reward Fine-Tuning (RFT) (Yuan et al., 2023) is another RL-based approach that demonstrates strong performance in mathematical reasoning tasks. Furthermore, Guided Reward Policy Optimization (GRPO) (Shao et al., 2024b) effectively improves the reasoning capabilities of LLMs without relying on external toolkits or voting mechanisms. DeepSeek-R1 (Guo et al., 2025), for example, leverages GRPO to fine-tune its model and achieves superior performance compared to existing methods. Inspired by these approaches, recent work (Chen et al., 2025) adopts similar fine-tuning strategies to enhance the reasoning capabilities of language and multimodal models.

## C VEHICLE KINEMATICS IN PHYSICS-GROUNDED REWARDS

The physical constraints of autonomous driving systems are deeply rooted in vehicle kinematics and passenger comfort principles. Vehicle kinematics governs the relationship between steering geometry, tire friction, and acceleration limits, ensuring that predicted trajectories adhere to the physical capabilities of the vehicle. For instance, abrupt steering adjustments can violate the minimum turning radius determined by the vehicle's wheelbase $L$ and maximum steering angle $\delta_{\max}$. The minimum turning radius $R_{\min}$ is defined as

$$R_{\min} = \frac{L}{\sin(\delta_{\max})}, \tag{9}$$

where $L$ is the distance between the front and rear axles (wheelbase), and $\delta_{\max}$ is the maximum achievable steering angle of the front wheels. Any trajectory requiring a tighter turn than $R_{\min}$ would be physically infeasible, leading to tire slippage or loss of control. Additionally, lateral acceleration $a_c$ during cornering must satisfy

$$a_c = \frac{v^2}{R} \leq \mu g, \tag{10}$$

where $v$ is the vehicle speed, $R$ is the turning radius, $\mu$ is the tire-road friction coefficient (typically $\mu \approx 0.8$), and $g$ is gravitational acceleration ($9.81\,\mathrm{m/s^2}$). Exceeding this threshold results in unsafe side-slip, particularly on low-friction surfaces like wet or icy roads. Beyond kinematic feasibility, passenger comfort is critically tied to smooth motion dynamics. Sudden changes in acceleration, known as jerk $j(t)$, directly impact rider experience. Jerk is defined as

$$j(t) = \frac{da(t)}{dt}, \tag{11}$$

where $a(t)$ is the instantaneous acceleration. Human tolerance for jerk is generally below $2.5\,\mathrm{m/s^3}$, and abrupt steering or acceleration adjustments (e.g., $\theta_j - \theta_{j-1}$ or $v_k - v_{k-1}$) can induce discomfort jerky motions. Furthermore, rapid maneuvers amplify vibrations in the vehicle suspension system, modeled as a second-order differential equation:

$$m\ddot{x} + c\dot{x} + kx = F(t), \tag{12}$$

where $m$ is the sprung mass (mass supported by the suspension), $c$ is the damping coefficient, $k$ is the spring stiffness, $x$ is the vertical displacement of the suspension, and $F(t)$ is external forces (e.g., centrifugal force during sharp turns). Excessive $F(t)$ due to abrupt motions overwhelms the suspension, increasing perceived jolts and reducing ride quality.

These principles are directly addressed in the physics-grounded reward framework. By penalizing abrupt changes in steering angle and velocity through temporal smoothness terms $r_{\text{tem}}$, the method ensures that trajectories remain within the vehicle's kinematic limits while minimizing jerk and vibration. This approach aligns with the experimental validation in the main text, where removing $r_{\text{tem}}$ led to a 26.3% increase in trajectory error, underscoring the necessity of balancing geometric accuracy with physical and physiological constraints.

## D    DETAILED PROMPTS TO GENERATE COT DATA

During the supervised fine-tuning (SFT) stage of AutoDrive-R², we designed a structured input prompt to generate high-quality chain-of-thought (CoT) data for the nuScenesR²-6K datasets. The prompt template is as follows:

```
### Prompt:
You are given an image, a driving-related question, and its answer.
    Generate a four-stage reasoning process with explicit mathematical
    modeling and self-validation. Engage in an internal dialogue using
    expressions such as 'let me think', 'wait', 'Hmm', 'oh, I see', 'let'
    s break it down', etc, or other natural language thought expressions.
     It's encouraged to include self-reflection or verification in the
    reasoning process.

### Input Format:
- System Instructions: {original_task}
- Past Vehicle Status: {original_information}
- Prediction Task: {original_problem}
- Answer: {original_solution}

### Output Format:
### 1. Visual Analysis
"Image analysis results:
- Vehicle's intended direction: Left turn (steering wheel angle: \theta
    rad)
- Obstacles ahead:
  * Car detected ahead (moving right/left/straight)
  * Pedestrian crossing road (left/right side)
- Traffic signal: signal_status detected (red / green / yellow)"

### 2. Motion Modeling
"Using historical data in Past Vehicle Status with N time points:
```
$$t_1 : [x = x_1, y = y_1], v = v_1 m/s, a = (a_{x_1}, a_{y_1}) m/s^2$$
$$t_2 : [x = x_2, y = y_2], v = v_2 m/s, a = (a_{x_2}, a_{y_2}) m/s^2$$
...
$$t_n : [x = x_n, y = y_n], v = v_n m/s, a = (a_{x_n}, a_{y_n}) m/s^2$$

```
Calculations:
- Average acceleration:
```
$$a_{x_{avg}} = (\sum a_{x_i})/N = a_{x_{avg}} m/s^2$$
$$a_{y_{avg}} = (\sum a_{y_i})/N = a_{y_{avg}} m/s^2$$
```
- Velocity prediction:
```
$$v_x = v_n + a_{x_{avg}} \times \delta_t = v_{t0} + a_{x_{avg}} \times \delta_t$$
$$v_y = v_n + a_{y_{avg}} \times \delta_t = v_{t0} + a_{y_{avg}} \times \delta_t$$
```
- Position prediction:
```
$$x(t+1) = x_n + v_x \times \delta_t + 0.5 \times a_{x_{avg}} \times \delta_t^2$$
$$y(t+1) = y_n + v_y \times \delta_t + 0.5 \times a_{y_{avg}} \times \delta_t^2$$
```
- Lateral offset: $\delta_y = v \times tan(\theta) = v_{t0} \times tan(\theta)$
```

```
### 3. Logical Deductions
"Safety check:
- If following this trajectory, will the vehicle:
  * Run a red light? → yes/no
  * Collide with car ahead? → yes/no
  * Hit pedestrian crossing? → yes/no
- Conclusion: recommended_action (e.g., 'Stop immediately', 'Reduce speed
    to 5m/s')"

### 4. Self-Reflection Validation
"Validation:
- Assumption check:
  * Predicted position (x=x_pred, y=y_pred) requires average speed of v m
      /s
  * Is this speed achievable with vehicle's acceleration history? → yes/
      no
- Adjustment:
  * If not feasible → Modify trajectory by reducing speed or increasing
      stopping distance"
```

This structured prompt ensures nuScenesR²-6K dataset contains diverse and causally plausible reasoning process, which are critical for cultivating the model's foundational perception and planning capabilities before RL fine-tuning.

## E  LLM CLARIFICATION

We clarify the role of Large Language Models (LLMs) in the preparation of this manuscript. Specifically, LLMs were employed to refine the language, grammar, and overall readability of the text. This involved tasks such as correcting grammatical errors, improving sentence structure, and enhancing the flow of the narrative to ensure clarity and precision in scientific communication. It is critical to emphasize that all core scientific contributions, including the formulation of the research problem, the development of the AutoDrive-R² framework, the creation of the nuScenesR²-6K dataset, the design of the two-stage training methodology combining chain-of-thought reasoning and Group Relative Policy Optimization (GRPO) with physics-grounded rewards, and the comprehensive experimental analysis on nuScenes and Waymo benchmarks, are the result of the authors' independent work. The LLM was strictly utilized as a linguistic tool and did not contribute to the conceptualization, implementation, or evaluation of the proposed methods. All intellectual property, including the theoretical foundations, algorithmic innovations, and empirical validations, originates solely from the human authors.

**"task"**: "You are an autonomous driving agent. You have access to a front view camera image of a vehicle <image>. Your task is to do your best to predict future waypoints for the vehicle over the next 3 timesteps, given the vehicle's intent inferred from the images."
**"information"**: "Provided are the previous ego vehicle status recorded over the last 0.5 seconds (at 0.5-second intervals). This includes the x and y coordinates of the ego vehicle. Positive x means forward direction while positive y means leftwards. The data is presented in the format [x, y]:.(t-0.5s) [-3.29, 0.04], Acceleration: X 0.3, Y 0.47 m/s^2, Velocity: 6.45 m/s, Steering angle: 0.33 (positive: left turn, negative: right turn), (t-0.0s) [0.0, 0.0], Acceleration: X 0.33, Y 0.59 m/s^2, Velocity: 6.53 m/s, Steering angle: 0.34 (positive: left turn, negative: right turn)"

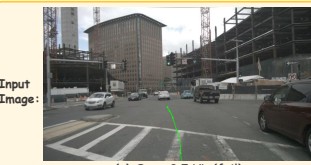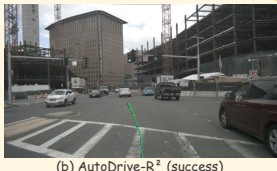

(a) Qwen2.5 VL (fail)  (b) AutoDrive-R² (success)

**Qwen2.5-VL-7B**: <think>Let me think through this step-by-step.### 1. Visual Analysis- **Vehicle's Intended Direction**: The vehicle is currently turning left, as indicated by the steering angle of 0.33 radians (positive for left turn).- **Obstacles Ahead**: There are cars ahead, but they appear to be moving in the same direction as the ego vehicle. No pedestrians are visible crossing the road.- **Traffic Signal**: The traffic light is green, allowing the vehicle to proceed.### 2. Motion Modeling- **Past Vehicle Status**: The vehicle has been turning left with a slight acceleration in both the x and y directions.- **Average Acceleration**: The average acceleration in the x-direction is approximately 0.33 m/s², and in the y-direction, it is approximately 0.59 m/s².- **Velocity Prediction**: Using the average acceleration, the velocity in the x-direction will be approximately 6.53 m/s + 0.33 m/s² × 0.5 s ≈ 6.66 m/s, and in the y-direction, it will be approximately 6.53 m/s + 0.59 m/s² × 0.5 s ≈ 6.77 m/s.- **Position Prediction**: The position in the x-direction will be approximately 0.0 + 6.66 m/s × 0.5 s + 0.33 m/s² × (0.5 s)² ≈ 3.42 m, and in the y-direction, it will be approximately 0.0 + 6.77 m/s × 0.5 s + 0.5 × 0.59 m/s² × (0.5 s)² ≈ 3.54 m.### 3. Logical Deductions- **Safety Check**: The vehicle is turning left and the traffic light is green, so there is no immediate danger of running a red light or colliding with the car ahead. The pedestrian crossing is not visible, so there is no risk of hitting a pedestrian.- **Recommended Action**: The vehicle should continue turning left at the current speed and trajectory.### 4. Self-Reflection Validation- **Assumption Check**: The predicted position (x ≈ 3.42 m, y ≈ 3.54 m) is feasible with the current acceleration history.- **Adjustment**: No adjustment is needed as the trajectory is safe and achievable.Given the analysis, the vehicle will continue turning left and proceed straight ahead.</think> <answer>[3.42, 3.54], [3.42, 3.54], [3.42, 3.54]</answer>",

**AutoDrive-R²**: <think>**1. visual analysis** \"image analysis results: - vehicle's intended direction: **left turn** (steering wheel angle: θ ≈ 0.34 rad) - obstacles ahead: **\* car detected ahead (moving straight) \* no pedestrian crossing road - traffic signal: green light detected (signal_status: green)**\" ### **2. motion modeling** \"using historical data in **past vehicle status** with 2 time points: t₁: [x=-3.29, y=0.04], v=6.45m/s, a=(0.3, 0.47)m/s² t₂: [x=0.0, y=0.0], v=6.53m/s, a=(0.33, 0.59)m/s² calculations: - average acceleration: a_avg_x = (0.3 + 0.33)/2 ≈ 0.315m/s² a_avg_y = (0.47 + 0.59)/2 ≈ 0.53m/s² - velocity prediction: v_x = 6.53 + 0.315 × 0.5 ≈ 6.66m/s v_y = 0 + 0.53 × 0.5 ≈ 0.265m/s - position prediction: x(t+1) = 0 + 6.66 × 0.5 + 0.5 × 0.315 × (0.5)² ≈ 3.33m y(t+1) = 0 + 0.265 × 0.5 + 0.5 × 0.53 × (0.5)² ≈ 0.16m - lateral offset: Δy = 6.53 × tan(0.34) ≈ 0.16m\" ### **3. logical deductions** \"safety check: - if following this trajectory, will the vehicle: \* run a red light? → no (traffic signal is green) \* collide with car ahead? → no (adequate distance maintained) \* hit pedestrian crossing? → no - conclusion: recommended_action (e.g., 'continue with current speed and trajectory')\" ### **4. self-reflection validation** \"validation: - assumption check: \* predicted position (x=3.33, y=0.16) requires average speed of 6.66m/s \* is this speed achievable with vehicle's acceleration history? → yes (vehicle has been accelerating steadily) - adjustment: \* if not feasible → modify trajectory by reducing speed or increasing stopping distance - oh, i see, the vehicle is on a clear path with no immediate obstacles, so the predicted trajectory seems valid. let's break it down further for the next steps.\" ### **final prediction** \"based on the analysis, the predicted future waypoints for the next 3 seconds are: [3.33, 0.05], [6.79, 0.16], [9.59, 0.38], [13.12, 0.75], [16.65, 1.22], [19.98, 1.68]\" </think> <answer>[3.33, 0.05], [6.79, 0.16], [9.59, 0.38], [13.12, 0.75], [16.65, 1.22], [19.98, 1.68]</answer>",

Figure 6: Visualization comparison between Qwen2.5-VL-7B and AutoDrive-R² on nuScenes.

**"task"**: "You are an autonomous driving agent. You have access to a front view camera image of a vehicle <image>. Your task is to do your best to predict future waypoints for the vehicle over the next 3 timesteps, given the vehicle's intent inferred from the images."
**"information"**: "Provided are the previous ego vehicle status recorded over the last 3.0 seconds (at 0.5-second intervals). This includes the x and y coordinates of the ego vehicle. Positive x means forward direction while positive y means leftwards. The data is presented in the format [x, y]:.(t-3.0s) [-17.85, -0.94], Acceleration: X 0.11, Y -0.26 m/s^2, Velocity: 5.92 m/s, Steering angle: 0.09 (positive: left turn, negative: right turn), (t-2.5s) [-14.9, -0.66], Acceleration: X 0.05, Y -0.45 m/s^2, Velocity: 5.92 m/s, Steering angle: -0.06 (positive: left turn, negative: right turn)...

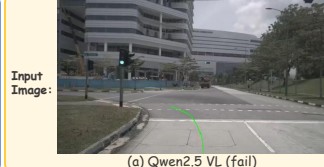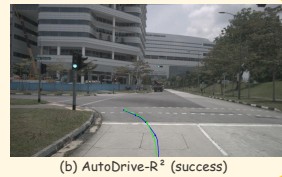

(a) Qwen2.5 VL (fail)  (b) AutoDrive-R² (success)

**Qwen2.5-VL-7B**: <think>Let me think through this step-by-step.### 1. Visual Analysis- **Vehicle's Intended Direction**: The steering angle is negative, indicating a left turn.- **Obstacles Ahead**: There is a car ahead moving straight, and a pedestrian is crossing the road on the left side.- **Traffic Signal**: The traffic signal is not visible in the image, so we cannot determine its status.### 2. Motion Modeling- **Historical Data**: The vehicle has been accelerating and turning left. The average acceleration can be calculated from the provided data.- **Average Acceleration**: - \\( a_{avg_x} = \\frac{1.15 + 1.16 + 1.36 + 1.31 + 1.2 + 1.16}{6} \\approx 1.22 \\), \\\\text{m/s}^2 \\) - \\( a_{avg_y} = \\frac{-1.86 - 1.66 - 1.17 - 0.49 - 0.3 - 0.22}{6} \\approx -1.13 \\, \\\\text{m/s}^2 \\)- **Velocity Prediction**: - \\( v_x = 5.59 + 1.22 \\times 0.5 \\approx 6.26 \\, \\\\text{m/s} \\) - \\( v_y = 5.59 - 1.13 \\times 0.5 \\approx 5.02 \\, \\\\text{m/s} \\)- **Position Prediction**: - \\( x(t+1) = 0.0 + 6.26 \\times 0.5 + 0.5 \\times 1.22 \\times 0.5^2 \\approx 3.53 \\, \\\\text{m} \\) - \\( y(t+1) = 0.0 + 5.02 \\times 0.5 + 0.5 \\times (-1.13) \\times (0.5)^2 \\approx 2.51 \\, \\\\text{m} \\)### 3. Logical Deductions- **Safety Check**: - The vehicle is turning left and accelerating. The pedestrian is crossing on the left, which could be a potential collision risk. - The car ahead is moving straight, so there is no immediate collision risk with it. - The traffic signal status is unknown, so we cannot determine if the vehicle will run a red light.### 4. Self-Reflection Validation- **Assumption Check**: - The predicted position (x=3.53, y=2.51) requires an average speed of approximately 6.26 m/s. - This speed is achievable with the vehicle's acceleration history, as the maximum speed in the data is 5.59 m/s, and the acceleration is positive.### Final PredictionBased on the calculations and the trajectory, the vehicle will continue to turn left and accelerate. The pedestrian crossing on the left could pose a risk, so the vehicle should be cautious.</think> <answer>[3.53, 2.51], [3.84, 2.76], [4.15, 3.01]</answer>",

**AutoDrive-R²**: <think>**1. visual analysis** \"image analysis results: - vehicle's intended direction: the vehicle appears to be preparing for a slight left turn or maintaining a straight path, as the steering angle is negative but small. **the road markings and traffic lights suggest a straight path ahead.** - obstacles ahead: \* no immediate obstacles like cars or pedestrians are visible in the image. the road appears clear. **\* the traffic light is green, indicating the vehicle can proceed.** - traffic signal: the traffic light is green, allowing the vehicle to continue moving forward.\"### **2. motion modeling** \"using historical data in **past vehicle status** with 7 time points: t₁: [x=-17.85, y=-0.94], v=5.92m/s, a=(0.11, -0.26)m/s² t₂: [x=-14.9, y=-0.66], v=5.92m/s, a=(0.05, -0.45)m/s² ... t₇: [x=0.0, y=0.0], v=5.78m/s, a=(-0.09, -0.38)m/s² calculations: - average acceleration: a_avg_x = (0.11 + 0.05 + 0.19 + 0.11 + 0.16 + 0.03 - 0.09)/7 ≈ 0.08m/s² a_avg_y = (-0.26 - 0.45 - 0.71 - 0.65 - 0.53 - 0.64 - 0.38)/7 ≈ -0.54m/s² - velocity prediction: v_x = 5.78 + 0.08 × 0.5 ≈ 5.82m/s v_y = 0 + (-0.54) × 0.5 ≈ -0.27m/s - position prediction: x(t+1) = 0 + 5.82 × 0.5 + 0.5 × 0.08 × (0.5)² ≈ 2.91m y(t+1) = 0 + (-0.27) × 0.5 + 0.5 × (-0.54) × (0.5)² ≈ -0.16m - lateral offset: Δy = v × tan(θ) = 5.78 × tan(-0.31) ≈ -0.33m\"### **3. logical deductions** \"safety check: - if following this trajectory, will the vehicle: \* run a red light? → no, the traffic light is green. \* collide with car ahead? → no, no cars are visible ahead. \* hit pedestrian crossing? → no, no pedestrians are visible. - conclusion: recommended_action: continue moving forward with slight adjustments for the left turn.\"### **4. self-reflection validation** \"validation: - assumption check: \* predicted position (x=2.91, y=-0.16) requires an average speed of approximately 5.82m/s. \* is this speed achievable with vehicle's acceleration history? → yes, the vehicle has been maintaining a similar speed. - adjustment: \* the trajectory seems feasible. however, the vehicle should continue monitoring the road for any unexpected obstacles or changes in traffic conditions.\" ### **final prediction** based on the analysis, the predicted future waypoints for the next 3 seconds are: <answer>[2.91, -0.06], [5.72, -0.15], [8.33, -0.15], [10.74, 0.19], [12.95, 0.76], [15.36, 1.66]</answer> the vehicle should continue moving forward with slight adjustments for the left turn, ensuring safety and adherence to traffic rules.</think> <answer>[2.91, -0.06], [5.72, -0.15], [8.33, -0.15], [10.74, 0.19], [12.95, 0.76], [15.36, 1.66]</answer>",

Figure 7: Visualization comparison between Qwen2.5-VL-7B and AutoDrive-R² on nuScenes.

