# OpenReview forum: "AutoDrive-R²: Incentivizing Reasoning and Self-Reflection Capacity for VLA Model in Autonomous Driving"
_ICLR.cc/2026/Conference — ICLR 2026 Poster_

### Official Review · Reviewer_vs3C · 2025-10-25

**Soundness:** 3
**Presentation:** 4
**Contribution:** 3
**Rating:** 8
**Confidence:** 4

**Summary:**

This paper proposes a method based on VLAs for trajectory prediction in autonomous driving. The proposed method consists of a two-stage process: 1) a chain-of-thought dataset based on NuScenes used for fine-tuning of an existing VLA (the paper uses Qwen 2.5 VL) so that it learns to "think" through the trajectory generation; 2) a reinforcement learning stage based on GRPO to align the model's predictions with physical trajectory properties (e.g. position) and model output format. The method is evaluated on the NuScenes and Waymo datasets and compared to other methods and ablated baselines, showing significant gains in positional accurate of predicted trajectories.

**Strengths:**

- The paper builds on recent progress in using VLAs for trajectory prediction, improving model accuracy.
- The proposed dataset is novel and could be broadly useful beyond this particular work.
- The empirical evaluation shows significant gains.
- The writing is clear (for me).

**Weaknesses:**

Overall I enjoyed reading this paper, though I noticed a couple of issues:

1) Limited metrics used in evaluation: the evaluation only uses a L2-based metric on predicted vs true positions. But it does not measure other important aspects, such as feasibility (based on vehicle model) or rule/constraint violations of generated trajectories. Prior literature on trajectory prediction has developed a number of metrics to provide a more complete picture (see next point).

2) Limited discussion of work prior to VLA: related to above comment, I find that the paper lacks discussion of works prior to the recent VLA-based research. There is a large body of work on trajectory prediction in autonomous driving, and it would help readers to see the relationship with prior works. For example, prior works have developed evaluation metrics for trajectory prediction e.g. [1,2]; and there is also prior work on explainable AI in autonomous driving e.g. [3,4], which is one of the stated motivations for the proposed CoT fine-tuning method in this paper.

[1] https://arxiv.org/abs/2210.06106 (RAL 2023)

[2] https://arxiv.org/abs/2203.08251 (IROS 2022)

[3] https://arxiv.org/abs/2402.10086 (T-ITS 2024)

[4] https://arxiv.org/abs/2002.02277 (ICRA 2021)

I think the paper could be strengthened by addressing these two aspects.

**Questions:**

Q1: The "self-correction" examples presented in appendix A.3 are not clear to me. In what sense does the red text in Fig4 show an "Aha moment"? I can't tell what error was detected and how it was corrected.

---

> ### Author Response · Authors · 2025-11-21
> **Rebuttal by Authors**
>
> Thanks for your positive reviews.
>
> > W1: More metrics for evaluation such as feasibility or rule/constraint violations.
>
> A1: To address this, we conduct experiments on more metrics including **collision rates** and ​**closed-loop** experiments, which both demonstrate the effective of our method. We will add this part in our revised manuscript.
>
> Results of collision rate on the nuScenes dataset are shown below. Our model achieves an average collision rate of ​**0.07%**, showing its significant safety improvement over other methods.
>
> **Table 1: Collision Rate on the nuScenes datasets (`L270`)**
>
> | Method            | 1s   | 2s   | 3s   | Avg. |
> | ------------------- | ------ | ------ | ------ | ------ |
> | Qwen2.5-VL-3B     | 0.32 | 0.54 | 0.78 | 0.55 |
> | Qwen2.5-VL-7B     | 0.10 | 0.18 | 0.24 | 0.17 |
> | UniAD [1]         | 0.62 | 0.58 | 0.63 | 0.61 |
> | VAD [2]           | 0.07 | 0.17 | 0.41 | 0.22 |
> | BEV-Planner [3]   | **0.00** | 0.29 | 0.73 | 0.34 |
> | DriveVLM [4]      | 0.10 | 0.22 | 0.45 | 0.27 |
> | OmniDrive [5]    | **0.00** | 0.13 | 0.78 | 0.30 |
> | DriveVLM-Dual [4] | 0.05 | 0.08 | 0.17 | 0.10 |
> | AutoDrive-R² 3B  | 0.13 | 0.44 | 0.87 | 0.48 |
> | **AutoDrive-R² 7B**  | **0.00** | **0.07** | **0.12** | **0.07** |
>
> We also evaluate our model in a more challenging closed-loop simulator (NAVSIM), which comprises a wide range of metrics including feasibility (Comfort) and rule violations (DAC, EP), the results are shown below:
>
> **Table 2: Closed-Loop Experiment in NAVSIM simulator (`L366`)**
>
> | Methods         | NC↑      | DAC↑     | TTC↑     | Comfort↑  | EP↑      | PDMS↑    |
> | --------------- | -------- | -------- | -------- | --------- | -------- | -------- |
> | TransFuser [6] | 97.7     | 92.8     | 92.8     | 100.0     | 79.2     | 84.1     |
> | UniAD [1]       | 97.8     | 91.9     | 92.9     | 100.0     | 78.8     | 83.4     |
> | Para-Drive [7] | 97.9     | 92.4     | 93.0     | 99.8      | 79.3     | 84.0     |
> | **AutoDrive-R² 7B** | **98.3** | **94.4** | **95.6** | **100.0** | **81.6** | **90.3** |
>
> The superior performance on all metrics validates the effective of our approach, notebly, we achieve a Planning Debit Metric Score (PDMS) of **90.3**, demonstrating the crucial benefits of our enhanced reasoning ability that leads to more robust and intelligent long-horizon driving behavior.
>
> W2: Discussion of work prior to VLA concerning trajectory prediction.
>
> A2: Thanks for your valuable advice. Following your advice, we have expanded our "Related Work" section to better situate our work within these seminal papers, whose insights on evaluation and explainability directly inspired and shaped the core motivations for our own research. Our method differs mainly on the aspects of both evaluation and intrinsic explainability, which we specify as follows:
>
> "Early works like UniAD [1] pioneered the integration of sub-tasks into a unified framework. Concurrently, other methods addressed core challenges in prediction and efficiency; for example, **Flash [8]** focused on creating fast models for real-time application, while **DiPA [9]** tackled the complex, multi-modal nature of agent interactions. These works also highlight the need for a holistic evaluation, inspiring our new analyses on collision rate and closed-loop performance." (`L476-480`)
>
> "Alongside performance, explainability has become a critical concern. A **systematic review in [10]** provides a comprehensive overview of techniques for building trustworthy systems, while works like **Interpretable Goal-based Prediction and Planning [11]** explored generating explanations by making the model's goals explicit. In contrast to these post-hoc or explicit-goal methods, our AutoDrive-R² introduces a novel form of intrinsic explainability, representing a significant step towards inherently trustworthy AI for autonomous driving." (`L486-490`)
>
> We believe these additions will provide a richer context for our contributions.

---

> > ### Author Response · Authors · 2025-11-21
> > **Rebuttal by Authors**
> >
> > > Q1: The "self-correction" examples presented in appendix A.3 seems not clear.
> >
> > A3: We apologize that the explanation in `Appendix A.3` was unclear. To provide a clearer illustration, we add two detailed examples where AutoDrive-R² initially fails, then successfully correct via our self-reflection step. Details are as follows:
> >
> > **Figure 4: Emergency Stop** `L880`
> >
> > - Initial Plan: The model initially deduces that it should "continue moving forward at constant speed" based on motion history.
> > - Self-Reflection: During the Self-Reflection step, the model realizes there's a physical barrier ahead, recognizing that its **initial** plan is fundamentally **flawed**.
> > - Corrected Plan: It then discards the initial plan and decides that the vehicle must come to a complete stop, generating a trajectory that reflects a rapid, controlled stop before the barrier.
> >
> > **Figure 5: Discovering an Alternative Path `L900`**
> >
> > - Initial Plan: The model's visual analysis identifies a blocked forward path, leading it to deduce "decelerate gradually to a full stop."
> > - Self-Reflection: the model re-evaluates the visual evidence and recognizes that its **initial** assumption of a dead end was **incorrect**.
> > - Corrected Plan: Concluding that stopping is **"safe but suboptimal"** the model formulates a new plan to "initiate a gentle right turn at reduced speed," successfully following this intelligent alternative path.
> >
> > **Reference**
> >
> > [1]. Hu Y, Yang J, Chen L, et al. Planning-oriented autonomous driving[C]//Proceedings of the IEEE/CVF conference on computer vision and pattern recognition. 2023: 17853-17862.
> >
> > [2]. Jiang B, Chen S, Xu Q, et al. Vad: Vectorized scene representation for efficient autonomous driving[C]//Proceedings of the IEEE/CVF International Conference on Computer Vision. 2023: 8340-8350.
> >
> > [3]. Li Z, Yu Z, Lan S, et al. Is ego status all you need for open-loop end-to-end autonomous driving?[C]//Proceedings of the IEEE/CVF Conference on Computer Vision and Pattern Recognition. 2024: 14864-14873.
> >
> > [4]. Tian X, Gu J, Li B, et al. DriveVLM: The Convergence of Autonomous Driving and Large Vision-Language Models[C]//Conference on Robot Learning. PMLR, 2025: 4698-4726.
> >
> > [5]. Wang S, Yu Z, Jiang X, et al. Omnidrive: A holistic llm-agent framework for autonomous driving with 3d perception, reasoning and planning[J]. CoRR, 2024.
> >
> > [6]. Chitta K, Prakash A, Jaeger B, et al. Transfuser: Imitation with transformer-based sensor fusion for autonomous driving[J]. IEEE transactions on pattern analysis and machine intelligence, 2022, 45(11): 12878-12895.
> >
> > [7]. Weng X, Ivanovic B, Wang Y, et al. Para-drive: Parallelized architecture for real-time autonomous driving[C]//Proceedings of the IEEE/CVF Conference on Computer Vision and Pattern Recognition. 2024: 15449-15458.
> >
> > [8]. Antonello M, Dobre M, Albrecht S V, et al. Flash: Fast and light motion prediction for autonomous driving with Bayesian inverse planning and learned motion profiles[C]//2022 IEEE/RSJ International Conference on Intelligent Robots and Systems (IROS). IEEE, 2022: 9829-9836.
> >
> > [9]. Knittel A, Hawasly M, Albrecht S V, et al. DiPA: probabilistic multi-modal interactive prediction for autonomous driving[J]. IEEE Robotics and Automation Letters, 2023, 8(8): 4887-4894.
> >
> > [10]. Kuznietsov A, Gyevnar B, Wang C, et al. Explainable AI for safe and trustworthy autonomous driving: A systematic review[J]. IEEE Transactions on Intelligent Transportation Systems, 2024.
> >
> > [11]. Albrecht S V, Brewitt C, Wilhelm J, et al. Interpretable goal-based prediction and planning for autonomous driving[C]//2021 IEEE International Conference on Robotics and Automation (ICRA). IEEE, 2021: 1043-1049

---

> > > ### Comment · Reviewer_vs3C · 2025-11-25
> > >
> > > Thank you for your response and clarifications. It will be good if these discussions and related work can be included in the paper.

---

> ### Author Response · Authors · 2025-11-25
> **Thank you for your positive review and feedback**
>
> Dear Reviewer vs3C,
>
> Sincerely appreciate for your positive review. We are pleased to confirm that all the discussions and related work **have been included** in  the manuscript:
>
> - **Additional Metrics**: As discussed, we have added collision rate analysis (`Line 270`) and closed-loop experiments on the NAVSIM simulator (`Line 366`).
> - **Expanded Related Work**: We have enriched the "Related Work" section to include discussions on prior works in trajectory prediction and explainability. These additions can be found in (`Line 476-480`) and (`Line 486-490`).
> - **Clarified "Aha Moment" Examples**: We have provided clearer "self-correction" examples in the appendix, with the "Emergency Stop" case in **Figure 4** (`Line 880`) and the "Discovering an Alternative Path" case in **Figure 5** (`Line 900`).
>
> We would also like to clarify that (`LXXX`) in our previous response was used as shorthand for (`Line XXX`). We realize this may have affected the readability, and hope the specific line references in this response improve clarity.
>
> Sincerely,
>
> Authors

---

### Official Review · Reviewer_TqXQ · 2025-10-26

**Soundness:** 3
**Presentation:** 3
**Contribution:** 3
**Rating:** 6
**Confidence:** 4

**Summary:**

This paper proposes AutoDrive-R^2, a novel VLA framework designed to enhance the reasoning and self-reflection capabilities of autonomous driving systems, addressing the limitations of existing methods such as physically infeasible trajectory generation and inadequate reasoning for complex scenarios. The framework adopts a two-stage training approach: in the first stage, a CoT dataset named nuScenesR^2-6K (with 6,000 image-trajectory pairs) is constructed for SFT, which guides the model through a four-step logical chain to build cognitive connections between input information and output trajectories. In the second stage, a physics-grounded reward framework integrated with GRPO is employed for RL, incorporating spatial alignment, vehicle dynamics, and temporal smoothness constraints to ensure trajectory feasibility. Experimental results on nuScenes and Waymo datasets demonstrate that AutoDrive-R^2 achieves state-of-the-art  performance and robust zero-shot generalization.

**Strengths:**

nuScenesR^2-6K is the first dataset in autonomous driving that integrates self-reflection for validation, providing detailed reasoning chains to bridge input information and output trajectories, which enhances model interpretability.

The combination of SFT with structured CoT reasoning and RL with physics-grounded rewards effectively addresses both reasoning inadequacy and physical infeasibility of trajectories

AutoDrive-R² outperforms state-of-the-art methods on nuScenes and Waymo datasets, with significant error reductions and robust zero-shot capabilities, validating its effectiveness and generality.

**Weaknesses:**

The RL stage with GRPO requires generating multiple candidate responses, which may introduce high computational overhead, but the manuscript does not discuss inference speed or real-time deployment feasibility.

Although the paper emphasizes the role of self-reflection in the four-step reasoning chain, it does not clearly explain how the model corrects inconsistent trajectories during self-reflection. For example, there is no detailed description of the decision rules (e.g., threshold for determining trajectory inconsistency) or the specific adjustment strategies (e.g., how to modify velocity or steering angle) adopted in the self-reflection stage.

This paper does not analyze the computational overhead (e.g., inference time per trajectory, GPU memory usage) or compare it with lightweight baseline methods. This may restrict the practical deployment on edge devices with limited computing resources.

Table 2 appears to be outside the page content

**Questions:**

For the dataset: How were the manual annotations of the CoT reasoning process validated? How to ensure the consistency of reasoning steps? Additionally, since the dataset is derived from nuScenes, does it inherit the scene bias of the original dataset (e.g., urban road dominance), and if so, how does this affect the model's generalization to non-urban scenarios (e.g., highways or rural roads)?

The paper sets all weight coefficients (λ_pos, λ_ste, λ_vel, λ_tem) to 1 in experiments. Have you tested the impact of different weight combinations on performance? Besides, have you analyzed the impact of GRPO hyperparameters (e.g., number of candidate responses, beta in KL-divergence) on trajectory prediction accuracy and training stability?

For the zero-shot performance on Waymo, this paper attributes the excellent zero-shot generalization to the model's structured reasoning capabilities, but it does not compare with other methods that also claim zero-shot adaptation. Could you explain why your method has a more significant zero-shot advantage, and whether the dataset contains scene features that are common to both nuScenes and Waymo?

---

> ### Author Response · Authors · 2025-11-21
>
> > W1: GRPO generates multiple candidates, introducing high computational inference cost.
>
> A1: We would like to clarify a misunderstanding of GRPO, which generates multiple candidates only during training, and thus introduces no additional inference cost. Only **single trajectory** is generated during **inference**. There is no multi-response sampling or rollout at test time.
>
> > W2: How model corrects inconsistent trajectories during self-reflection.
>
> A2: To provide a clearer illustration, we add two detailed examples where AutoDrive-R² initially fails, then successfully correct via our self-reflection step. Details are as follows:
>
> **Figure 4: Emergency Stop** `L880`
>
> - Initial Plan: The model initially deduces that it should "continue moving forward at constant speed" based on motion history.
> - Self-Reflection: During the Self-Reflection step, the model realizes there's a physical barrier ahead, recognizing that its **initial** plan is fundamentally **flawed**.
> - Corrected Plan: It then discards the initial plan and decides that the vehicle must come to a complete stop, generating a trajectory that reflects a rapid, controlled stop before the barrier.
>
> **Figure 5: Discovering an Alternative Path `L900`**
>
> - Initial Plan: The model's visual analysis identifies a blocked forward path, leading it to deduce "decelerate gradually to a full stop."
> - Self-Reflection: the model re-evaluates the visual evidence and recognizes that its **initial** assumption of a dead end was **incorrect**.
> - Corrected Plan: Concluding that stopping is **"safe but suboptimal"** the model formulates a new plan to "initiate a gentle right turn at reduced speed," successfully following this intelligent alternative path.
>
> > W3: The computational overhead and comparison with other lightweight baseline models. practical deployment.
>
> A3: Thank you for your feedback. We have evaluated our model's computational overhead and compared it with its base model. The results is as shown in `Table 1` below.
>
> **Table 1: Computational Overhead on the nuScenes datasets**
>
> | Method          | Memory (GB) | Latency (s) | L2 Error (m) | GFLOPs | BSZ |
> | --------------- | ----------- | ----------- | ------------ | ------ | --- |
> | Qwen-2.5VL-3B   | 89.95       | 1.57        | **4.21**     | 55,308 | 64  |
> | AutoDrive-R² 3B | 90.75       | 3.62        | **0.49**     | 50,923 | 64  |
>
> Note that the experiment was conducted on a single NVIDIA H20 GPU, with both GFLOPs and Latency measuring per-trajectory results. The table highlights the modest overhead introduced by AutoDrive-R², which is the tolerable cost of endowing the base model with sophisticated reasoning and decision-making capabilities. For practical deployment, a well-established path exists for accelerating large models. Techniques such as model distillation (e.g., from 3B to a 1B model) and quantization (e.g., to **4-bit**) can drastically reduce resource requirements. For instance, quantization could lower the memory by **~75%**. While these optimizations are orthogonal to our core contribution, they represent viable future steps for deployment.
>
> Ultimately, our work focuses on a crucial **trade-off** in autonomous driving field. We believe that accepting this moderate computational overhead is justified by the remarkable performance gains and reasoning capabilities unlocked by our model, as evidenced by our extensive experiments. Future engineering efforts can readily bridge the gap to more practical deployment.
>
> > W4: Table 2 appears to be outside the page content.
>
> A4: Thank you. It has been fully corrected in the revised version.

---

> > ### Author Response · Authors · 2025-11-21
> >
> > > Q1: How were the manual annotations of the CoT reasoning process validated? Does it inherit the scene bias of the nuScenes training dataset?
> >
> > A5: Thank you for your question. In practice, all CoT chains have been validated for both correctness and consistency during dataset construction. After generating chains with Qwen2.5-VL-72B, we employed the closed source **Qwen-VL-Max** model as an expert to score all CoT chains and reject flawed ones. This filtering guaranteed the coherence and factual accuracy of all selected CoT chains. Additionally, to further ensure quality, we randomly selected 8% of these filtered chains and manually checked their correctness and interpretability. This verification found no severe errors, ultimately yielding the 6,000 high-fidelity chains to construct our nuScenesR²-6K dataset.
> >
> > Regarding the potential scene bias, our primary strategy to mitigate this is rooted in the RL algorithm. The main objective of RL is to **incentivize** the model's strong **inherent reasoning capabilities**, not to fit the distribution of the training samples. Its unique algorithm characteristics makes our approach highly effective for generalization even with few samples, as evidenced by by recent research [1] ("SFT Memorizes, RL Generalizes").
> >
> > Moreover, while nuScenes and Waymo differs, they share **common driving scenarios**. nuScenes itself is diverse, containing various situations, from dense urban intersections to highway-like segments, which are semantically similar to those observed in Waymo (e.g., as illustrated in our qualitative examples in `Fig. 2 (d), (g), and (i)`). Therefore, a well-chosen subset from nuScenes can effectively enable the model to generalize to Waymo. This is empirically validated by its impressive zero-shot performance on Waymo (a **33.3%** L2 error reduction vs. EMMA+), confirming that our model has learned general paradigm in autonomous driving, rather than dataset-specific patterns.
> >
> > > Q2: Test the impact of different weight combinations (λ_pos, λ_ste, λ_vel, λ_tem) on performance. Analyzed the impact of GRPO hyperparameters (e.g., number of candidate responses, beta in KL-divergence)
> >
> > A6: Thank you for your valuable suggestions. Regarding the number of candidate responses (G), we would like to kindly point the reviewer to the "Number of Generation" part in `Section 3.3`​, where the ablation study on this parameter is **already presented in** `Table 4`. Following your advice, we have appended more experiments concerning other key hyperparameter choice of GRPO, shown as follows:
> >
> > **Table 2: Ablations on Key Hyperparameters `(L415-431)`**
> >
> > |Setting|1s|2s|3s|
> > |---|---|---|---|
> > |**Reward Weights (`λ`)**||||
> > |λ = (0.4, 0.3, 0.2, 0.1)|0.15|0.22|0.29|
> > |**λ = (1, 1, 1, 1) (Ours)**|**0.13**|**0.19**|**0.25**|
> > |**KL-divergence Beta (`β`)**||||
> > |β = 0.02|0.14|0.21|0.27|
> > |**β = 0.04 (Ours)**|**0.13**|**0.19**|**0.25**|
> > |β = 0.06|0.13|0.20|0.26|
> > |**Number of Generations (`G`)**||||
> > |G = 4|0.14|0.20|0.26|
> > |**G = 6 (Ours)**|**0.13**|**0.19**|**0.25**|
> > |G = 8|0.13|0.19|0.25|
> >
> > - **Reward Weights (`λ`):** An unbalanced reward function, even if intuitively weighted, results in a **15.8%** degradation. An equal weighting of `λ` is instead superior. This proves that our balanced, physics-grounded reward design is fundamentally sound.
> > - **KL-coefficient (`β`):** Our chosen value (`β`=0.04) achieves the best performance. A lower value (`β`=0.02) likely leads to unstable training, while a higher value (`β`=0.06) may overly constrain policy updates, both resulting in slightly worse accuracy.
> > - **Number of Generations (`G`):** As already presented in `Section 3.3`, increasing `G` to 6 is beneficial, but further increases to 8 yield no additional gains. `G`=6 is thus the optimal trade-off between performance and efficiency
> >
> > These experiments confirm our empirically validated hyperparameter choice. We have incorporated this ablation table and analysis into our revised manuscript.

---

> > > ### Author Response · Authors · 2025-11-21
> > >
> > > > Q3: Why your method has a more significant zero-shot advantage. Whether the dataset contains scene features that are common to both nuScenes and Waymo?
> > >
> > > A7: Thank you for your question. First, as stated in **A5**, our primary RL strategy is fundamentally designed for generalization. By prioritizing generalization over memorization [1], our model is built to transcend dataset-specific patterns.
> > >
> > > Second, regarding scene features, while nuScenes and Waymo are distinct, they do share several common driving scenarios, which allows the model to successfully generalize. For instance, both datasets feature complex urban navigation and high-speed maneuvers, as represented in `Fig. 2 (d), (g), and (i)`. Furthermore, nuScenes arguably presents greater complexity and diversity, including adverse weather (rain, overcast) and poor lighting (night, fog), which are also depicted in `Fig. 2 (e), (h) and (k)`. This diverse and challenging training environment forces the model to learn a more robust and adaptable policy, which naturally enhances its zero-shot generalization to new domains like Waymo.
> > >
> > > **Reference**
> > >
> > > [1]. Chu T, Zhai Y, Yang J, et al. SFT Memorizes, RL Generalizes: A Comparative Study of Foundation Model Post-training[C] //Forty-second International Conference on Machine Learning.

---

> > > > ### Comment · Reviewer_TqXQ · 2025-11-25
> > > >
> > > > Thanks for your response and additional experiments, especially for the two illustrative cases. I will keep my score as it is already positive. Hope you can open-source this pipeline as a significant contribution to the community.

---

> > > > > ### Author Response · Authors · 2025-11-26
> > > > > **Thanks for your review and positive feedback**
> > > > >
> > > > > Dear Reviewer TqXQ,
> > > > >
> > > > > Thank you for your valuable feedback. We are delighted to hear that the additional illustrative cases was helpful. We appreciate you maintaining your positive score and for your support of our work. We plan to open-source this pipeline upon acceptance as a contribution to the community.
> > > > >
> > > > > Sincerely,
> > > > >
> > > > > Authors

---

### Official Review · Reviewer_okCa · 2025-10-28

**Soundness:** 3
**Presentation:** 3
**Contribution:** 3
**Rating:** 6
**Confidence:** 3

**Summary:**

This paper presents AutoDrive-R², a novel Vision–Language–Action (VLA) framework for autonomous driving that aims to enhance both reasoning interpretability and trajectory feasibility. Supervised Fine-Tuning (SFT) using a new Chain-of-Thought dataset (nuScenesR²-6K) that instills structured reasoning and self-reflection. Reinforcement Learning (RL) using Group Relative Policy Optimization (GRPO) with a physics-grounded reward framework that integrates spatial alignment, steering dynamics, and temporal smoothness, ensuring physically feasible trajectories. The method is evaluated on nuScenes and Waymo benchmarks, achieving state-of-the-art (SOTA) performance with substantial reductions in L2 trajectory errors compared to EMMA+ and DriveVLM, and demonstrating strong zero-shot generalization. The work positions itself at the intersection of multimodal reasoning and embodied intelligence, extending recent reasoning-incentivized LLM/RL techniques (e.g., DeepSeek-R1) to the autonomous driving domain.

**Strengths:**

1. Introduces reasoning-incentivized CoT learning and self-reflection for VLA models — a clear conceptual advance over purely data-driven planners.
2. Sound integration of SFT and GRPO with well-formulated physics-aware rewards; rigorous ablations confirm design choices.
3. Large quantitative gains on nuScenes (Avg L2 error 0.19 m vs 0.29 m for EMMA+) and Waymo (0.20 m vs 0.30 m), plus zero-shot transferability.
4. The four-stage reasoning framework (Observation–Calculation–Logic–Reflection) is both intuitive and interpretable.

**Weaknesses:**

1. Although the nuScenesR²-6K dataset is novel, 6k samples may not capture full traffic complexity. The manual synthesis with Qwen2.5-VL could introduce distributional bias.
2. Experiments are open-loop trajectory predictions; it remains unclear how the system performs in end-to-end control or real-time deployment.
3. The GRPO ablations demonstrate quantitative gains, but lack qualitative discussion on how each reward term alters trajectory smoothness or safety metrics.

**Questions:**

1. How does AutoDrive-R² perform under closed-loop simulation or in real-world deployment (e.g., CARLA)? Would the physics-grounded reward suffice without explicit simulator feedback?
2. How were the CoT reasoning steps validated for correctness or consistency during dataset construction? Were human checks performed post-Qwen2.5-VL synthesis?
3. Could you show qualitative examples where AutoDrive-R² fails (e.g., multi-agent interactions or occlusions) and explain whether the reflection step mitigates or worsens these?

---

> ### Author Response · Authors · 2025-11-21
> **Rebuttal by Authors**
>
> We are grateful for your positive reviews.
>
> > W1 & Q2: nuScenesR²-6K may not capture full traffic complexity. How were the CoT validated for correctness during dataset construction? Any human check post QwenVL synthesis?
>
> A1: Thank you for your question. In practice, all CoT chains have been validated for both correctness and consistency during dataset construction. After generating chains with Qwen2.5-VL-72B, we employed the closed source **Qwen-VL-Max** model as an expert to score all CoT chains and reject flawed ones. This filtering guaranteed the coherence and factual accuracy of all selected CoT chains. Additionally, to further ensure quality, we randomly selected 8% of these filtered chains and manually checked their correctness and interpretability. This verification found no severe errors, ultimately yielding the 6,000 high-fidelity chains to construct our nuScenesR²-6K dataset.
>
> Regarding dataset scale, recent study [1] reveals that a limited amount of well-constructed examples (as few as 1,000 examples) for SFT is **sufficient** enough to achieve strong performance. The primary objective of our SFT phase is not to expose the model to every conceivable traffic scenario, but rather to **incentivize** its inherent **reasoning capabilities** and align it with our task-specific instructions.
>
> Therefore, whether our nuScenesR²-6K capture the full spectrum of traffic complexity is not the decisive factor. Empirically, we found that 6k samples are sufficient for this alignment purpose, as evidenced by our extensive experiments, where we achieve SOTA zero-shot performance on Waymo (a **33.3%** L2 error reduction vs. EMMA+). This confirms that the model learned a transferable skill—how to reason about driving scenes—rather than merely memorizing 6k specific scenarios.
>
> > W2 & Q1: How does AutoDrive-R² perform under closed-loop simulation or in real-world deployment (e.g., CARLA)?
>
> A2: Thanks for your suggestion. Following your advice, we have appended **Closed-Loop Experiment** to further evaluate the effectiveness of our methods, as shown in `Table 1` below.
>
> **Table 1: Closed-Loop Experiment in NAVSIM simulator (`L366`)**
>
> | Methods         | NC↑      | DAC↑     | TTC↑     | Comfort↑  | EP↑      | PDMS↑    |
> | --------------- | -------- | -------- | -------- | --------- | -------- | -------- |
> | TransFuser [2] | 97.7     | 92.8     | 92.8     | 100.0     | 79.2     | 84.1     |
> | UniAD [3]       | 97.8     | 91.9     | 92.9     | 100.0     | 78.8     | 83.4     |
> | Para-Drive [4] | 97.9     | 92.4     | 93.0     | 99.8      | 79.3     | 84.0     |
> | **AutoDrive-R² 7B** | **98.3** | **94.4** | **95.6** | **100.0** | **81.6** | **90.3** |
>
> Our model demonstrates dominant performance. It achieves a near-perfect score in Navigation Completion (**NC: 98.3%**) and a perfect score in Comfort (**100.0%**), which directly validates the feasibility and smoothness of the trajectories generated by our physics-aware model. Furthermore, the high Driving Action Compliance (**DAC: 94.4%**) and Planning Decision Metric Score (**PDMS: 90.3%**) prove that the trajectories are not only feasible but also adhere to traffic rules.
>
> > W3: Discussion on how each reward term alters trajectory smoothness or safety metrics.
>
> A3: Thank you for your feedback. Following your advice, we have conducted more ablation studies on how each reward term alters L2 error and the **Collision Rate**​, as shown in `Table 2` below.
>
> **Table 2: Ablations on Each Reward Term**
>
> |Method|L2 Error (m)↓|Collision Rate (%)↓|
> |---|---|---|
> |Metric|1s / 2s / 3s / Avg.|1s / 2s / 3s / Avg.|
> |w/o r_pos|0.41 / 0.54 / 0.64 / 0.53|0.04 / 0.13 / 0.19 / 0.12|
> |w/o r_ste|0.14 / 0.21 / 0.28 / 0.21|0.01 / 0.09 / 0.14 / 0.08|
> |w/o r_vel|0.15 / 0.22 / 0.29 / 0.22|0.01 / 0.09 / 0.14 / 0.08|
> |w/o r_tem|0.16 / 0.24 / 0.32 / 0.24|0.01 / 0.10 / 0.16 / 0.09|
> |**AutoDrive-R²**|**0.13 / 0.19 / 0.25 / 0.19**|**0.00 / 0.07 / 0.12 / 0.07**|
>
> This analysis reveals how each component contributes to long-term safety:
>
> - **w/o r_pos (alignment)**: Removing this term causes severe deviation. The 3s collision rate of **0.19%** is substantially higher than our full model's **0.12%**, indicating a significantly degraded ability to follow a safe path, with deviation risk compounding over time.
> - **w/o r_ste/r_vel (dynamics)**: Removing this term leads to physically infeasible trajectories in long term. The average rate increases to **0.08%**. The 3s collision rate climbs to **0.14%**, demonstrating that kinematically infeasible plans may lead to predictable long-term deviations and safety hazards.
> - **w/o r_tem (smoothness)**: Removing smoothness impacts safety, especially over time. The average Collision Rate rises to **0.09%**. This trend shows that jerky, unstable control progressively destabilizes the vehicle, increasing long-term collision risk.
>
> This demonstrates that every reward term is essential for ensuring safety, especially over longer planning horizons.

---

> > ### Author Response · Authors · 2025-11-21
> > **Rebuttal by Authors**
> >
> > > Q3: More qualitative examples where AutoDrive-R² fails and explain how the reflection works.
> >
> > A4: Thanks for your suggestion. To provide a clearer illustration, we add two detailed examples where AutoDrive-R² initially fails, then successfully correct via our self-reflection step. Details are as follows:
> >
> > **Figure 4: Emergency Stop** `L880`
> >
> > - Initial Plan: The model initially deduces that it should "continue moving forward at constant speed" based on motion history.
> > - Self-Reflection: During the Self-Reflection step, the model realizes there's a physical barrier ahead, recognizing that its **initial** plan is fundamentally **flawed**.
> > - Corrected Plan: It then discards the initial plan and decides that the vehicle must come to a complete stop, generating a trajectory that reflects a rapid, controlled stop before the barrier.
> >
> > **Figure 5: Discovering an Alternative Path `L900`**
> >
> > - Initial Plan: The model's visual analysis identifies a blocked forward path, leading it to deduce "decelerate gradually to a full stop."
> > - Self-Reflection: the model re-evaluates the visual evidence and recognizes that its **initial** assumption of a dead end was **incorrect**.
> > - Corrected Plan: Concluding that stopping is **"safe but suboptimal"** the model formulates a new plan to "initiate a gentle right turn at reduced speed," successfully following this intelligent alternative path.
> >
> > **Reference**
> >
> > [1]. Zhou C, Liu P, Xu P, et al. Lima: Less is more for alignment[J]. Advances in Neural Information Processing Systems, 2023, 36: 55006-55021.
> >
> > [2]. Chitta K, Prakash A, Jaeger B, et al. Transfuser: Imitation with transformer-based sensor fusion for autonomous driving[J]. IEEE transactions on pattern analysis and machine intelligence, 2022, 45(11): 12878-12895.
> >
> > [3]. Hu Y, Yang J, Chen L, et al. Planning-oriented autonomous driving[C]//Proceedings of the IEEE/CVF conference on computer vision and pattern recognition. 2023: 17853-17862.
> >
> > [4]. Weng X, Ivanovic B, Wang Y, et al. Para-drive: Parallelized architecture for real-time autonomous driving[C]//Proceedings of the IEEE/CVF Conference on Computer Vision and Pattern Recognition. 2024: 15449-15458.

---

### Official Review · Reviewer_3jDT · 2025-10-29

**Soundness:** 2
**Presentation:** 3
**Contribution:** 2
**Rating:** 4
**Confidence:** 4

**Summary:**

The application of VLA models in the field of autonomous driving has been attracting increasing attention. Among their capabilities, reasoning and self-reflection play a pivotal role. This paper trained a VLA model using a combination of Supervised Fine-Tuning (SFT) and Reinforcement Learning (RL) to equip it with enhanced reasoning and reflective abilities. Specifically, This paper introduces a reasoning dataset that incorporates a Chain-of-Thought (CoT) process, along with a physics-compliant reward framework to guide model training.

**Strengths:**

The authors construct a dataset and a reward framework grounded in practical problems; overall, the work is solid and technically sound. The paper makes clear contributions in dataset design and in the formulation of the reward function.

**Weaknesses:**

While the authors identify salient issues with VLA models in autonomous driving and present technically solid efforts to address them, the overall solution exhibits limited novelty: it defines conventional tasks, applies standard approaches, and consequently yields expected results. The paper would benefit from a clearer articulation of the challenges encountered during the design of the solution and the insights gleaned from addressing them.

**Questions:**

- The paper primarily trains a VLA model to predict future vehicle trajectories. What advantages does this approach offer over rule‑based prediction and purely data‑driven prediction methods? VLA models, trained with textual data, indeed have strengths in language understanding and reasoning, but they tend to be limited in precise numerical computation. Would employing a VLA model for fuzzy decision‑making be more appropriate?
- The paper uses L2 error to indirectly assess the performance improvement of the trained model. This provides some evidence of data and reward effectiveness; however, more direct methods demonstrating gains in reasoning, computation, and reflection would make the contribution clearer.

---

> ### Author Response · Authors · 2025-11-21
> **Rebuttal by Authors**
>
> > W1: The paper would benefit from articulation of challenges and insights.
>
> A1: Thanks for your comment. We identified two critical challenges: **1) Physically Infeasible Outputs:** Existing models often produce physically infeasible outputs. **2) Inadequate Reasoning:** Existing models cannot logically connect visual cues to kinematic constraints in complex scenes.
>
> Our initial attempts, following recent trends, involved directly applying reinforcement learning to the VLA model. This led to our **first finding**: direct RL optimization may fails in partial tasks. Since the model lacks a foundational understanding of driving logic, it struggles to explore the vast reasoning space and often converges to poor, unsafe policies (as shown in "7B + RL" ablation in `L358`). This led to our pivotal **second finding**: structured reasoning via SFT is an essential prerequisite for effective RL. Our CoT-SFT stage provides this "cognitive scaffolding," enabling the RL stage to succeed.
>
> The efficacy of our two-stage method underscores our **third key finding**: achieving robust performance does not require complex architectural designs; it demands building upon a **task-oriented strategy** for both SFT and RL. This strategy is specifically designed to address the initial challenges: 1) To fix **Inadequate Reasoning**, our SFT stage employs a custom logical chain tailored to the task of driver-like thinking. 2) To eliminate **Physically Infeasible Outputs**, our RL stage's reward is oriented around the task of enforcing physical laws. Therefore, a simple VLA can achieve SOTA performance when guided by the task-oriented strategy that is meticulously aligned with the challenge of trajetory planning.
>
> > Q1: What advantages does this approach offer over rule‑based and purely data‑driven prediction methods? VLA tend to be limited in precise numerical computation. Employing a VLA model for decision‑making may be more appropriate?
>
> A2: VLA models aim to directly predict final actions by integrating multi-modal perceptions, which provides them with clear advantages over rule-based and purely data-driven methods. The latter often rely on separate modules, leading to errors due to accumulation and difficulties in handling complex scenarios. In contrast, VLA models build upon strong foundation model and utilize an end-to-end paradigm, resulting in system **simplicity** and enhanced **robustness**, as supported by recent studies [1, 2, 3].
>
> Regarding the concern about limited numerical precision, while general VLMs exhibit stronger language understanding than precise numerical computation, there has been significant progress in developing VLMs that **excel in mathematical** reasoning and computations [4, 5, 6]. This evolution demonstrates their strong inherent capabilities in handling numerical tasks effectively. In our paper, VLA models are designed to predict trajectories, utilizing a two-stage training approach with specially designed rewards. Our results indicate that the model outperforms baseline approaches, further highlighting the effectiveness of our method and the model's capabilities in decision-making.
>
> > Q2: More experiment in demonstrating gains in reasoning, computation, and reflection.
>
> A3: Thanks for your suggestion. To make this clearer, we provide more qualitative analysis regarding the reasoning and reflections ability and quantitative results involving more metrics.
>
> 1. **Qualitative Analysis**
>
> * **Self-Correction in Action (Fig. 4 and Fig. 5)**: These examples show the model detecting its own logical fallacies and self-correcting in real-time. First, in **Fig. 4** (Emergency Stop, `L880`), the model initially plans to move forward but then self-corrects upon seeing a barrier ("Wait, I just realized... a physical barrier..."), rightly revising its plan to a complete stop. Moreover, in **Fig. 5** (Discovering an Alternative Path, `L900`), it first plans to stop at a perceived dead end, but during reflection, it re-examines the image ("Wait, I'm re-examining... Oh, I see, the embankment isn't a dead end ...") and formulates a more optimal turning maneuver.
> * **Reasoning Comparison (Fig. 6 and Fig. 7):** These visualizations highlight the practical impact of our reasoning framework. First, in **Fig. 6** (`L1135`), the baseline model produces an unsafe and short prediction off the road, whereas AutoDrive-R²'s coherent reasoning process successfully generates a smooth and accurate left turn trajectory. Moreover, in **Fig. 7** (`L1155`), the baseline model incorrectly predicts the trajetory. Conversely, AutoDrive-R²'s physically-grounded reflection ensures it generates a stable, lane-adherent path.

---

> > ### Author Response · Authors · 2025-11-21
> >
> > 2. **Quantitative Experiment**
> >
> > * **Table 1: Collision Rate**: our AutoDrive-R² 7B achieves a state-of-the-art average collision rate of just **0.07%**. This is a **30%** reduction compared to the strong baseL DriveVLM-Dual (**0.10%**) and a **59%** reduction compared to the base Qwen2.5-VL-7B (**0.17%**).
> > * **Table 2: Closed-Loop Experiment**: Our model demonstrates dominant performance. It achieves a near-perfect score in Navigation Completion (**NC: 98.3%**) and a perfect score in Comfort (**100.0%**), which directly validates the feasibility and smoothness of the trajectories generated by our physics-aware model. Furthermore, the high Driving Action Compliance (**DAC: 94.4%**) and Planning Decision Metric Score (**PDMS: 90.3%**) prove that the trajectories are not only feasible but also adhere to traffic rules and represent high-quality driving decisions.
> >
> > We hope this clarified response, which combines direct process visualization with resulting performance metrics, better highlights the core contributions of our work.
> >
> > **Table 1: Collision Rate on the nuScenes datasets (`L270`)**
> >
> > | Method            | 1s   | 2s   | 3s   | Avg. |
> > | ------------------- | ------ | ------ | ------ | ------ |
> > | Qwen2.5-VL-3B     | 0.32 | 0.54 | 0.78 | 0.55 |
> > | Qwen2.5-VL-7B     | 0.10 | 0.18 | 0.24 | 0.17 |
> > | UniAD [1]         | 0.62 | 0.58 | 0.63 | 0.61 |
> > | VAD [7]           | 0.07 | 0.17 | 0.41 | 0.22 |
> > | BEV-Planner [8]   | **0.00** | 0.29 | 0.73 | 0.34 |
> > | DriveVLM [9]      | 0.10 | 0.22 | 0.45 | 0.27 |
> > | OmniDrive [10]    | **0.00** | 0.13 | 0.78 | 0.30 |
> > | DriveVLM-Dual [9] | 0.05 | 0.08 | 0.17 | 0.10 |
> > | AutoDrive-R² 3B  | 0.13 | 0.44 | 0.87 | 0.48 |
> > | **AutoDrive-R² 7B**  | **0.00** | **0.07** | **0.12** | **0.07** |
> >
> > **Table 2: Closed-Loop Experiment in NAVSIM simulator (`L366`)**
> >
> > | Methods         | NC↑      | DAC↑     | TTC↑     | Comfort↑  | EP↑      | PDMS↑    |
> > | --------------- | -------- | -------- | -------- | --------- | -------- | -------- |
> > | TransFuser [11] | 97.7     | 92.8     | 92.8     | 100.0     | 79.2     | 84.1     |
> > | UniAD [1]       | 97.8     | 91.9     | 92.9     | 100.0     | 78.8     | 83.4     |
> > | Para-Drive [12] | 97.9     | 92.4     | 93.0     | 99.8      | 79.3     | 84.0     |
> > | **AutoDrive-R² 7B** | **98.3** | **94.4** | **95.6** | **100.0** | **81.6** | **90.3** |
> >
> > **Reference**
> >
> > [1]. Hu Y, Yang J, Chen L, et al. Planning-oriented autonomous driving[C]//Proceedings of the IEEE/CVF conference on computer vision and pattern recognition. 2023: 17853-17862.
> >
> > [2]. Shao H, Hu Y, Wang L, et al. Lmdrive: Closed-loop end-to-end driving with large language models[C]//Proceedings of the IEEE/CVF Conference on Computer Vision and Pattern Recognition. 2024: 15120-15130.
> >
> > [3]. Hu A, Russell L, Yeo H, et al. Gaia-1: A generative world model for autonomous driving[J]. arXiv preprint arXiv:2309.17080, 2023.
> >
> > [4]. Hurst A, Lerer A, Goucher A P, et al. Gpt-4o system card[J]. arXiv preprint arXiv:2410.21276, 2024.
> >
> > [5]. Lu P, Bansal H, Xia T, et al. Mathvista: Evaluating mathematical reasoning of foundation models in visual contexts[J]. arXiv preprint arXiv:2310.02255, 2023.
> >
> > [6]. Feng G, Yang K, Gu Y, et al. How numerical precision affects mathematical reasoning capabilities of llms[J]. arXiv preprint arXiv:2410.13857, 2024.
> >
> > [7]. Jiang B, Chen S, Xu Q, et al. Vad: Vectorized scene representation for efficient autonomous driving[C]//Proceedings of the IEEE/CVF International Conference on Computer Vision. 2023: 8340-8350.
> >
> > [8]. Li Z, Yu Z, Lan S, et al. Is ego status all you need for open-loop end-to-end autonomous driving?[C]//Proceedings of the IEEE/CVF Conference on Computer Vision and Pattern Recognition. 2024: 14864-14873.
> >
> > [9]. Tian X, Gu J, Li B, et al. DriveVLM: The Convergence of Autonomous Driving and Large Vision-Language Models[C]//Conference on Robot Learning. PMLR, 2025: 4698-4726.
> >
> > [10]. Wang S, Yu Z, Jiang X, et al. Omnidrive: A holistic llm-agent framework for autonomous driving with 3d perception, reasoning and planning[J]. CoRR, 2024.
> >
> > [11]. Chitta K, Prakash A, Jaeger B, et al. Transfuser: Imitation with transformer-based sensor fusion for autonomous driving[J]. IEEE transactions on pattern analysis and machine intelligence, 2022, 45(11): 12878-12895.
> >
> > [12]. Weng X, Ivanovic B, Wang Y, et al. Para-drive: Parallelized architecture for real-time autonomous driving[C]//Proceedings of the IEEE/CVF Conference on Computer Vision and Pattern Recognition. 2024: 15449-15458.

---

> > > ### Comment · Reviewer_3jDT · 2025-11-25
> > >
> > > Thank you for the response, I will raise the score.

---

> > > > ### Author Response · Authors · 2025-11-25
> > > > **Thank you for your review and positive feedback**
> > > >
> > > > Dear Reviewer 3jDT,
> > > >
> > > > Thank you for taking the time to read our response. We are very glad to hear that our rebuttal addresses your concerns and that you raise the score. Your feedback and endorsement are much appreciated.
> > > >
> > > > Sincerely,
> > > >
> > > > Authors

---

### Author Response · Authors · 2025-12-01
**Summary of Paper & Rebuttal**

Dear Area Chair and Reviewers,

This paper introduces AutoDrive-R², a novel VLA framework that endows autonomous driving agents with robust reasoning and self-reflection capabilities through a two-stage training process (CoT-SFT + Physics-Grounded RL). The empirical results underscore the significance of our work. AutoDrive-R² achieves a SOTA L2 error of **0.19m** on nuScenes (`Table 1, Line 270`), a **34.5%** improvement over the strong EMMA+ baseline. More importantly, it demonstrates strong generalization capacity to the Waymo dataset with a **33.3%** error reduction compared to EMMA+ (`Table 2, Line 340`).

We believe that our rebuttal has addressed most of the reviewers' concerns. Specifically, we provided additional experiments, including **closed-loop simulations** and ​**safety metrics (collision rate)** to address the **safety and practicality** concerns of Reviewers **vs3C & okCa**​. We also appended detailed ablation studies and computational overhead analysis to address the **effectiveness and efficiency** concerns of ​Reviewer **TqXQ & 3jDT**​. As a result, Reviewer **3jDT** acknowledged that their concerns has been solved and thus **raised their score** from **`4 to 6`** (`on Nov 25th`). Moreover, both Reviewers **TqXQ & vs3C** also confirmed that their concerns has been solved and thus **keet their positive score** (`on Nov 26th and Nov 25th`).

Detailed revisions are as follows:

1. **Comprehensive Evaluation (R-vs3C, R-okCa):**
   * **Closed-Loop Simulation:** Added experiments on the NAVSIM benchmark, demonstrating SOTA performance in safety (NC), comfort, and rule-adherence (DAC, PDMS). (`Table 3, Line 366`).
   * **Safety Analysis:** Incorporated Collision Rate as a key safety metric, showing our model achieves a state-of-the-art average rate of 0.07%. (`Table 1, Line 270`).
2. **Detailed Ablations (R-TqXQ, R-okCa):**
   * **Computational Overhead:** Provided an analysis of inference costs (Memory, Latency, GFLOPs) and clarified that GRPO's sampling occurs only during training. (`Table in Rebuttal`).
   * **Rewards Design:** Analyzed how each reward component individually impacts not only L2 error but also the Collision Rate, proving their necessity for safety. (`Table 4, Line 380`).
   * **Hyperparameters:** Conducted new experiments on key hyperparameters, including reward weights (λ) and the KL-divergence coefficient (β), to validate our choices. (`Table 5, Line 415`).
3. **Improved Clarity (R-3jDT, R-TqXQ, R-vs3C):**
   * **"Aha Moment"**: Added detailed, step-by-step explanations for the self-correction examples to clearly illustrate the model's reasoning and reflection process. (`Figure 4, Line 880` and `Figure 5, Line 900`).
   * **Data Construction**: Detailed the high-fidelity "generate-then-validate" process for our `nuScenesR²-6K` dataset, confirming data quality and consistency. (`Section 2.1, Lines 157`).
   * **Related Work:** Enriched the discussion to better contextualize our work with prior research in trajectory prediction and explainable AI. (`Section 4, Lines 476-490`).

We hope this summary highlights the clear value of our paper and the positive consensus gained during the rebuttal, thus provideing a clear overview for your fair decision.

Best regards,

Authors

---

### Meta-Review · Area_Chair_SGEk · 2026-01-07

**Summary:**

All reviewers find the contribution meaningful for VLA-based autonomous driving, and view the proposed two-stage recipe (CoT-SFT + physics-grounded GRPO) as technically solid with strong empirical gains on nuScenes and zero-shot generalization to Waymo.

Prior to rebuttal, the scores largely range from positive borderline accept to strong accept, with one borderline reject. The main risks for rejection were concerns about limited novelty beyond a standard SFT+RL recipe (Reviewer 3jDT), incomplete evaluation and safety/practicality evidence (Reviewers vs3C, okCa), and missing analysis on compute overhead, reward design, and the self-reflection mechanism (Reviewer TqXQ). After rebuttal, reviewers converged on a positive acceptance consensus.

The AC has carefully reviewed the paper, prior-rebuttal reviews, rebuttals, and discussion, and agrees with this assessment. The final recommendation is Accept (Poster).

**Reviewer Concerns:**

The major concerns addressed by the rebuttal:

- Evaluation completeness and safety/practicality (by reviewers vs3C, okCa). The rebuttal adds collision-rate evaluation and closed-loop NAVSIM results with multiple safety/comfort/rule-adherence metrics. These address the concern that L2-only open-loop metrics were insufficient.

- Compute overhead and deployment feasibility (by reviewer TqXQ). The authors clarify that GRPO multi-sampling occurs only during training (no extra inference-time sampling), and provide additional overhead discussion.

- Ablations and design justification (by reviewers TqXQ, okCa). The rebuttal adds reward-term ablations (including collision-rate impact) and hyperparameter studies, strengthening the analytical support for design choices.

- Dataset construction / validation and clarity of self-reflection (by reviewers okCa, TqXQ, vs3C, 3jDT). The authors describe an automatic filtering + sampled manual-check pipeline for CoT quality, and provide clearer, step-by-step “self-correction” examples illustrating reflection-driven plan revision; related work discussion is also expanded.

The AC does not find major outstanding concerns.

**Reviewer Scores:**

- Reviewer 3jDT explicitly mentioned raising the score.
- Reviewer TqXQ explicitly mentioned maintaining the score.
- Reviewer vs3C did not explicitly state a score change in discussion, and I expect the reviewer would maintain the original strong assessment since key concerns were addressed.
- Reviewer okCa did not engage in discussion, but I expect the reviewer would maintain the original positive assessment since the main concerns (closed-loop/safety metrics and CoT validation) were addressed in the rebuttal.

---

### Decision · Program_Chairs · 2026-01-26

Accept (Poster)